

# Simulating Wildfire Emissions and Plumerise using Geostationary Satellite Fire Radiative Power Measurements: A Case Study of the 2019 Williams Flats fire

Aditya Kumar[1*], R. Bradley Pierce[1], Ravan Ahmadov[2,3], Gabriel Pereira[4], Saulo Freitas[5], Georg Grell[3], Chris Schmidt[1], Allen Lenzen[1], Joshua P. Schwarz[6], Anne E. Perring[7], Joseph M. Katich[7,8], John Hair[9], Jose L. Jimenez[2, 10], Pedro Campuzano-Jost[2, 10], Hongyu Guo[2,10]

[1]Space Science and Engineering Center, University of Wisconsin Madison, Madison, WI

[2]Cooperative Institute for Research in Environmental Sciences (CIRES), University of Colorado Boulder, Boulder, CO

[3]NOAA Global Systems Laboratory, Boulder, CO

[4]Federal University of São João del-Rei, MG, Brazil

[5]Center for Weather Forecast and Climatic Studies (CPTEC), Brazil

[6]National Oceanic and Atmospheric Administration Chemical Sciences Laboratory, Boulder, CO

[7]Department of Chemistry, Colgate University, Hamilton, NY

[8]Now at Ball Aerospace, Boulder, CO

[9]National Aeronautics and Space Administration (NASA) Langley Research Center, Hampton, VA

[10]Department of Chemistry, University of Colorado Boulder, Boulder, CO

[*]Correspondence to Aditya Kumar (akumar98@wisc.edu)



# Abstract

We use the Weather Research and Forecasting with Chemistry (WRF-Chem) model with new implementations of GOES-16 fire radiative power (FRP) based wildfire emissions and plume-rise to interpret aerosol observations during the 2019 NASA-NOAA FIREX-AQ field campaign and perform model evaluations. We compare simulated aerosol concentrations and optical properties against observations of black carbon aerosol from the NOAA Single Particle Soot Photometer (NOAA-SP2), organic aerosol from the CU High Resolution Aerosol Mass Spectrometer (HR-AMS) and aerosol backscatter coefficients from the High Spectral Resolution Lidar (HSRL) system. This study focuses on the Williams Flats fire in Washington, which was repeatedly sampled during four science flights by the NASA DC-8 (August 3 – August 8, 2019). The emissions and plume-rise methodologies are implemented following NOAA's operational High Resolution Rapid Refresh coupled with Smoke (HRRR-Smoke) forecasting model. In addition, new GOES-16 FRP based diurnal cycle functions are developed and incorporated in WRF-Chem. The FIREX-AQ observations represented a diverse set of sampled environments ranging from fresh/aged smoke from the Williams Flats fire to remnants of plumes transported over long distances. The Williams Flats fire resulted in significant aerosol enhancements during August 3-8, 2019, which were substantially underestimated by the standard version of WRF-Chem. The simulated BC and OC concentrations increased between 92 – 125 times (BC) and 28-78 times (OC) with the new implementation compared to the standard WRF-Chem version. These increases resulted in better agreement with the FIREX-AQ airborne observations for BC and OC concentrations (particularly for fresh smoke sampling phases) and aerosol backscatter coefficients. The model still showed a low bias in simulating the aerosol loadings observed in aged plumes from Williams Flats. WRF-Chem with the FRP-based plumerise simulated similar plume heights



to the standard plumerise model in WRF-Chem. The simulated plume heights (for both versions)
compared well with estimated plume heights using the HSRL measurements. Therefore, the
improvements in the model simulation were mainly driven by the higher emissions in the FRP-
based version. The model evaluations also highlighted the importance of accurately accounting for
the wildfire diurnal cycle and including adequate representation of the underlying chemical
mechanisms, both of which could significantly impact model forecasting performance.













# 1. Introduction

Wildfires are episodic ecosystem disturbances that play a key role in shaping and overall functioning of terrestrial ecosystems (Bond et al., 2005;Pausas and Ribeiro, 2017) and provide several ecosystem services (Pausas and Keeley, 2019). They also emit large amounts of pollutants into the atmosphere which can have important implications for air quality (McClure and Jaffe, 2018;Jaffe et al., 2020), atmospheric chemistry/composition (Xu et al., 2021), human health (Xu et al., 2020), and the Earth's radiation budget (Jiang et al., 2020). A particular concern associated with wildfire events arises from the serious health effects wildfire smoke can have (e.g. (Reid et al., 2016)). Wildfire regimes have altered significantly over the past few years in the United States (US) with climate change hypothesized to be a major driving force (Flannigan et al., 2000;Holden et al., 2018;Halofsky et al., 2020). These alterations have been predicted to continue in the coming decades (e.g., Pechony and Shindell (2010)) resulting in growing concerns over the potential health impacts. In addition, long-range transport of smoke is a cause of concern for downwind communities.

Air quality forecasts generated by computational models are useful to assess the impacts a wildfire event could have on air quality (in the vicinity of the fire as well as at far away locations) and consequently the risk posed on human health due to smoke exposure. Thus, the accuracy of air quality forecasts both during fire events and in general is of paramount importance as highlighted by previous studies (e.g., Kumar et al. (2018);Al-Saadi et al. (2005)). Computational models used to provide air quality forecasts rely on a continuous ingestion of fire detections and properties available from either polar-orbiting or geostationary satellites and are run with the latest available information to generate smoke forecasts for the next few days (typically 36 to 72 hours). There are



several forecasting systems that have these models as a basis. Recently, Ye et al. (2021) have
discussed and evaluated these forecasting systems during the **F**ire **I**nfluence on **R**egional to **G**lobal
**E**nvironment and **A**ir **Q**uality (FIREX-AQ) field campaign in detail. The ability of computational
models to accurately simulate air quality impacts during wildfire events is critically dependent on
the inputs such as the estimated emissions, and the simulated altitude of the emissions (smoke
injection height, or plume-rise) (Val Martin et al., 2012;Carter et al., 2020).
Wildfire emissions in the past have primarily been estimated following the model of Seiler and
Crutzen (1980). There have been several fire emission inventories compiled over the years which
use this methodology as the fundamental basis (e.g., **G**lobal **F**ire **E**missions **D**atabase (GFED)
(Van Der Werf et al., 2004;2006;2010;2017), **F**ire **IN**ventory from the **N**ational Center for
Atmospheric Research (FINN) (Wiedinmyer et al., 2011)). However, this method is prone to
uncertainties given the large number of parameters involved (burned area estimates, available
biomass density, combustion efficiencies). Significant advances have been made in estimating the
burned area with refined global estimates available. However, the uncertainties associated with
available biomass density (ABD) and combustion efficiency estimates are particularly large and
persist (e.g., (Reid et al., 2009)). An alternative emissions estimation approach is based on using
the remote-sensing measurements of fire radiative power (FRP) and has formed the basis of
multiple recent emission inventories (e.g., **G**lobal **F**ire **A**ssimilation **S**ystem (GFAS) (Kaiser et al.,
2012), **Q**uick **F**ire **E**missions **D**ataset (QFED) (Darmenov and da Silva, 2015)). The major
advantage of this approach is a more direct estimation of fire emissions without the need to use a
multitude of parameters. In addition, Wiggins et al. (2020) found significant correlations between
GOES-16 FRP and in-situ measurements of important smoke tracers (e.g., $CO_2$, CO). Wiggins et



al. (2021) discuss in detail the differences in the two approaches to estimate fire emissions and the
underlying uncertainties.
In contrast to fire emission inventories, the issue of estimating plume-rise in computational models
has received considerably less attention. There have been a few plume rise approaches developed
in the past with a detailed list provided by Val Martin et al. (2012). The approach developed by
Freitas et al. (2007) (updates in Freitas et al. (2010)) has been the most commonly used. It has been
evaluated by past studies (e.g., (Val Martin et al., 2012)) and has been embedded in several
computational models including the **W**eather **R**esearch and **F**orecasting with **C**hemistry (WRF-
Chem) model (described in Section 2). In recent work, a modified version of this approach has
been included in the **H**igh- **R**esolution **R**apid **R**efresh coupled with **S**moke (HRRR-Smoke)
forecasting model (described in Section 3) run operationally at the **N**ational **O**ceanic and
**A**tmospheric **A**dministration (NOAA). The modified plume-rise approach incorporates FRP in
computing the plume-rise. HRRR-Smoke also includes an FRP-based approach to estimate fire
emissions. However, the HRRR-Smoke FRP-based approaches of estimating emissions and
plume-rise together with GOES-16 FRP measurements have not been implemented in other
computational models and no previous studies exist focusing on field observations based
evaluation of the performance in WRF-Chem.
The 2019 FIREX-AQ field campaign (Roberts et al., 2018) was jointly led by the **N**ational
**A**eronautics **S**pace **A**dministration (NASA) and NOAA. The campaign took place during July –
September 2019 in two phases. The first phase was held out of Boise (ID) (Figure 1 (a)) in the
Western US ((July – August 2019) referred to as phase 1 hereon) and the second phase was out of
Salina (KS) (Figure 1(b)) ((August – September 2019) referred to as phase 2 hereon) in the South-
Eastern US.




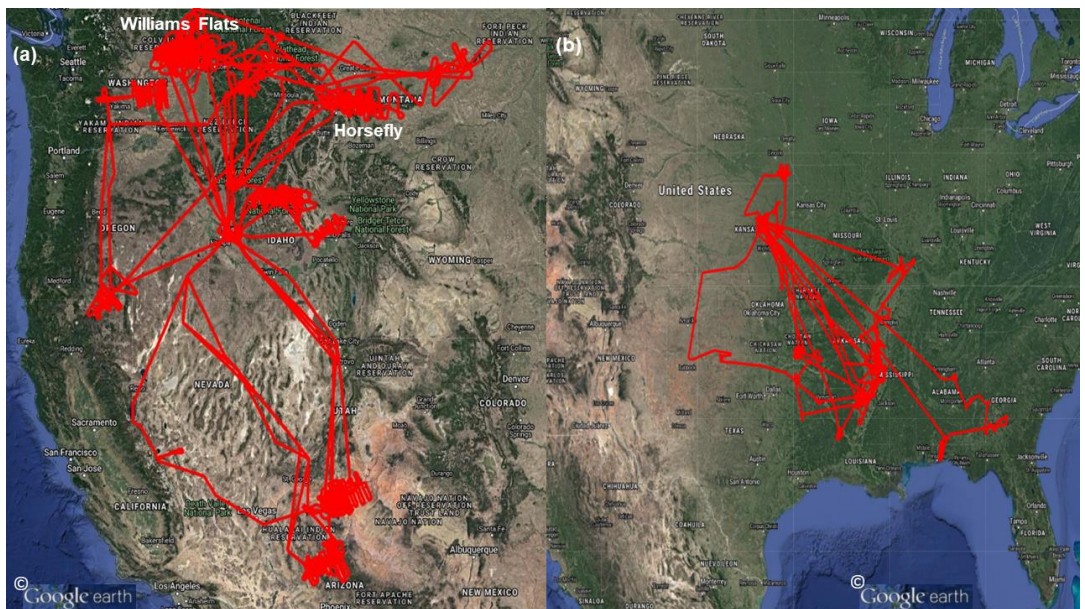


**Figure 1: NASA DC-8 flight tracks during the Boise phase (a, left) and Salina phase (b, right) of the 2019 FIREX-AQ field campaign. The locations of Williams Flats fire and Horsefly fire which are the main focus of this study are shown (in white). Image: © Google Earth**


Phase 1 focused on wildfires primarily in the Western U.S. while Phase 2 was aimed at sampling

agricultural (and prescribed) fires in the South-Eastern U.S. The campaign included a suite of

measurement platforms aimed at sampling fire smoke at different altitudes and different times of

the day. The goal of the campaign was to improve the current scientific understanding of fire

behavior, fire smoke chemistry and its impact on atmospheric composition and air quality.

Multiple airborne (NASA DC-8, NASA ER-2, NOAA CHEM-Twin Otter and NOAA MET-Twin

Otter) and ground based measurement platforms were employed during the campaign to get a

comprehensive sampling of the fires of interest. Mobile ground-based platforms (e.g., Aerodyne,

NASA Langley Mobile Laboratory) provided high resolution ground level sampling of fire smoke.

Wildfires occurring in different ecosystems and meteorological conditions and agricultural fires

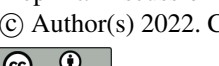



involving burning of different crop types were sampled using a suite of instruments aboard the
different aircrafts. High temporal resolution measurements (typically 1 Hz, up to 20 Hz for some
sensors) of important trace gas species (e.g., CO, $O_3$, $NO_x$, VOCs) and aerosols (e.g., BC, OC)
were carried out aboard the different aircraft. High Spectral Resolution Lidar (HSRL)
measurements of aerosol optical properties are also available for all DC-8 flights of the campaign.
This study uses the WRF-Chem model with FRP-based fire emissions and plume-rise estimation
methodologies employed in the HRRR-Smoke forecasting system to interpret aerosol observations
during the FIREX-AQ field campaign and perform evaluations of retrospective aerosol forecasts
with in-situ measurements available from the FIREX-AQ field campaign. Section 2 of this paper
provides a general overview of the modeling tools including the WRF-Chem model together with
details about the specific version being run at the University of Wisconsin Madison Space Science
and Engineering Center (UW Madison SSEC) and the HRRR-Smoke model. Section 3 describes
the data products used in this study including the GOES-16 fire product and in-situ measurement
data available from FIREX-AQ. Section 4 presents discussion/interpretation of the FIREX-AQ
observations and results from the model evaluation for the respective FIREX-AQ DC-8 science
flights.









## 2. Methodology

### 2.1. The WRF-Chem model

The WRF-Chem model (Grell et al., 2005) is a model of meteorology, atmospheric chemistry/physics, and transport. It builds on the existing WRF model (Skamarock et al., 2019;Powers et al., 2017), which is primarily a weather forecasting model, by including full coupling of the meteorological component with a chemistry component. WRF-Chem uses the **A**dvanced **R**esearch **W**RF (ARW) dynamical core to solve the flux-form of the non-hydrostatic Euler equations. It uses the Arakawa Staggered C-Grid horizontally whereas the vertical levels in the model are defined using a terrain following sigma-hybrid coordinate system. The **W**RF **P**reprocessing **S**ystem (WPS) is the input pre-processing component of WRF-Chem. It is used to pre-process the terrestrial (e.g., 2-D vegetation, soil data) and meteorological (e.g., 3-D temperature, pressure fields) data to be compatible with the WRF-Chem configuration (model domain extent, grid size etc.). The chemistry component includes emissions of atmospheric species (anthropogenic, biogenic, geogenic (dust and volcanoes), fires), chemical mechanisms for gas-phase species and aerosols and atmospheric loss processes. Each chemical mechanism can either be coupled with aerosol schemes or run by itself. Dry deposition parameterization in the model follows the resistance-based scheme of Wesely (1989). The model supports both 1-way and 2-way horizontal nesting. WRF-Chem includes several schemes for microphysics (e.g., WRF Single-Moment 3-Class (WSM3), Thompson etc.), surface layer, deep/shallow cumulus parameterization, land surface, planetary boundary layer, and atmospheric radiation.



## 2.2. WRF-Chem at University of Wisconsin Madison

We use the WRF-Chem version run in real-time at the University of Wisconsin Madison (WRFv3.5.1 and referred to as WRF-Chem hereon). It is a 1-way nested version of WRF-Chem and comprises of a regional domain spanning the continental United States (CONUS) with a horizontal spatial resolution of 8km and 34 vertical layers (Greenwald et al., 2016). This model is used to provide daily chemical forecasts (currently for aerosols only) over CONUS and was one of the participating models providing chemical forecasting assistance for flight planning during FIREX-AQ. It uses the **G**oddard **C**hemistry **A**erosol **R**adiation and **T**ransport/Georgia Tech-Goddard **G**lobal **O**zone **C**hemistry **A**erosol **R**adiation and **T**ransport (GOCART) mechanism to simulate tropospheric aerosol components (Chin et al., 2000a;2000b;2002;Ginoux et al., 2001). The simulated aerosol components include sulfate ($SO_4^{2-}$), hydrophilic and hydrophobic organic (OC) and black carbon (BC), dust, and sea-salt (SS) with no secondary organic aerosol (SOA) formation. No size distributions are included for $SO_4^{2-}$, OC and BC while a sectional scheme is used for dust (0.5, 1.4, 2.4, 4.5, 8.0 µm and SS (0.3, 1.0, 3.2, 7.5 µm). GOCART uses an OA/OC ratio of 1.8, which is generally appropriate for fresh biomass burning organic aerosol emissions (Andreae, 2019) but low for more aged aerosol (Hodzic et al., 2020). The Aerosol Optical Depth (AOD) in the model is calculated at 550 nm by vertical integration of the aerosol extinction. Hygroscopic growth is accounted for and extinction efficiencies are used as a function of mole fraction. The microphysics scheme is from Thompson et al. (2004), a modified version of the Rapid Radiative Transfer Model radiative scheme (RRTMG) is used for both shortwave (RRTMG_SW) and longwave (RRTMG_LW) radiation along with the Noah Land Surface Model (Noah-LSM) and the Mellor-Yamada-Janjic (Eta) surface layer scheme.





The initial (ICs) and lateral boundary conditions (LBCs) for meteorology and aerosol species (SO$_2$,
SO$_4^{2-}$,Dimethyl sulfide (DMS), BC, OC, dust, SS) are from the **G**lobal **F**orecast **S**ystem (GFS) and
the global component of the **R**ealtime **A**ir **Q**uality **M**odeling **S**ystem (referred to as RAQMS
hereon) (Pierce et al., 2003;2007;Natarajan et al., 2012) respectively. RAQMS combines chemical
modeling and assimilation to provide 4-day global chemical forecasts. The version providing
chemical ICs/LBCs for this study uses the GOCART mechanism, fire detections from MODIS,
has a spatial resolution of 1° x 1° and the University of Wisconsin (UW) hybrid isentropic
coordinate model as the dynamical core (Schaack et al., 2004). It has 35 vertical levels extending
from the surface to the upper stratosphere (terrain-following at the surface to isentropic in the
stratosphere). The modeling system is initialized with assimilation of total column ozone from the
**O**zone **M**onitoring **I**nstrument (OMI), ozone profiles from MLS and AOD from MODIS. It also
includes comprehensive stratospheric and tropospheric chemistry mechanisms (Pierce et al.,
2007), which have been extensively evaluated.
WRF-Chem employs the PREP-Chem (v1.3) emissions preprocessor (Freitas et al., 2011) to
compute daily emissions of atmospheric species. These emissions include anthropogenic, fires,
volcanic, and biogenic sources, which are input to WRF-Chem at the start of a simulation. Fire
emissions are based on the Brazilian Biomass Burning Emission Model (3BEM) (Longo et al.,
2010), which is a fire burned area based bottom-up approach. The original version of the model
was designed to use remote-sensing observations from both geostationary and polar-orbiting
satellites. The geostationary satellite data was from the GOES WF_ABBA product which included
the instantaneous fire size whereas for polar orbiting satellites a mean fire size was assumed. The
details of this approach are provided in Freitas et al. (2011). 3BEM computes daily emissions for
110 species for each fire location. PREP-Chem at UW Madison has been modified to use only the


GOES-16 Fire Detection and Characterization (FDC) product (described in Section 3.1). The
GOES-16 FDC algorithm is an extension of the GOES Wildfire Automated Biomass Burning
Algorithm (Section 3.1). Aboveground carbon density estimates are based on Olson et al. (2000)
with later updates by (Gibbs, 2006;2007). The land cover data (Belward, 1996;Sestini et al., 2003)
has a 1 km spatial resolution and 17 land cover types based on the **I**nternational **G**eosphere-
**B**iosphere **P**rogram (IGBP) land cover classification. Combustion factors and emission factors are
based on look up tables. Emission factors are from Andreae and Merlet (2001) and Longo et al.
(2009). The plume-rise model (Freitas et al., 2007;2010) is embedded in WRF-Chem and is a 1-
D time-dependent entrainment plume model. This model is used to simulate the vertical
distribution of emissions/plumerise for each WRF-Chem grid cell with a fire. It takes as input the
emissions for the grid cell, fire properties (e.g., fire size), and other parameters (e.g., meteorology,
land cover). The model provides as output the lower and upper levels between which the emissions
are to be distributed. PREP-Chem computes daily emissions for each fire location, aggregates them
on the 8km x 8km WRF-Chem grid and provides them as input (together with fire properties (e.g.,
fire size)) for WRF-Chem and its plumerise model which distributes the emissions in the vertical
domain. The diurnal cycle of wildfire emissions is simulated by using an analytical function which
peaks at 18Z (Figure 2). This is the default diurnal cycle available with WRF-Chem and was
developed based on fires in the Amazon (Freitas et al., 2011).
In operational/forecast mode, the model provides a 60-hour forecast every day. The forecast runs
are initialized at 0000 UTC and use fire detection and meteorology data from the previous day.
Fires are assumed to persist throughout the forecasting period. For this study, WRF-Chem was run
for 36-hour periods in retrospective mode with a specific focus on the Boise phase of the FIREX-
AQ field campaign.

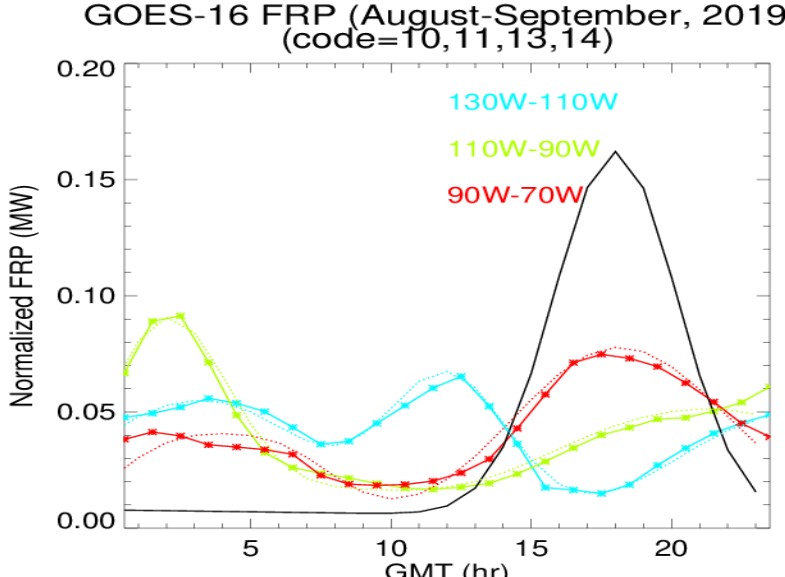


**Figure 2: The diurnal cycle functions (solid lines (green, blue, and red)) developed based on GOES-16 FRP data during the FIREX-AQ period. The original WRF-Chem diurnal cycle function is also shown (solid black line). The dashed lines (green, blue and red) show the normalized FRP.**

In retrospective mode, the model has the same configuration as the forecast mode except that fire
detections are for the current day, and meteorological data and initial/lateral boundary conditions
are from analyses. The modeling experiments consisted of two sets of simulations with different
WRF-Chem versions. Set 1 included the WRF-Chem version with the default PREP-Chem v1.3
fire emissions estimates, the Freitas et al. (2007) plumerise model described earlier in this section
(referred to as the 3BEM version hereon), and the diurnal cycle function peaking at 18Z. Set 2
included the version with fire radiative power (FRP) based emissions estimates and plumerise
model (referred to as FRP version hereon). The FRP based updates are implemented following the
**H**igh **R**esolution **R**apid **R**efresh **S**moke (HRRR-Smoke) modeling system which is a forecasting
modeling system providing high temporal and spatial resolution (3 km) smoke forecasts for



CONUS (using the VIIRS fire product) (described in the next section). We also developed new
diurnal cycle functions (solid red, blue, and green curves in Figure 2) by adapting the default
analytical function (shown in black in Figure 2) to match the mean diurnal GOES-16 FRP profiles
within three different longitude bands over the FIREX-AQ period (August-September 2019).
These diurnal functions were used in the FRP version.

## 2.3. HRRR-Smoke model


The **H**igh **R**esolution **R**apid **R**efresh **S**moke (HRRR-Smoke) model is a 3-D forecasting model
(https://rapidrefresh.noaa.gov/hrrr/HRRRsmoke/), which is run at NOAA/NCEP. It uses a single
smoke tracer to simulate smoke emissions and transport at a high spatial and temporal resolution
to provide real-time smoke forecasts. The model domain spans the CONUS with a horizontal
spatial resolution of 3 km and 50 vertical levels. HRRR-Smoke forecasts are initialized every hour
using the HRRR meteorological analyses with the forecast lead times varying between 18-48
hours. HRRR-Smoke is a coupled model where the direct radiative effects of smoke feedback on
the dynamics. The model uses fire location (latitude, longitude) and FRP measurements from 4
polar orbiting satellites, 2 VIIRS (375m resolution I-band Active Fire (AF) algorithm which is
based on the Moderate Resolution Imaging Spectroradiometer (MODIS) Collection 6 retrieval
(Giglio et al., 2016)) and 2 MODIS. It employs an FRP based methodology to estimate fire smoke
emissions and simulate plume-rise in the model. Smoke emissions in HRRR-Smoke are estimated
by using FRP measurements to derive the fire radiative energy (FRE) over the fire duration
(Ahmadov et al., 2017). The biomass burned is estimated by multiplying the FRE estimates with
conversion coefficients from Kaiser et al. (2012). The model accounts for variation in these





coefficients across ecosystems by using ecosystem specific conversion coefficients. The land
cover types in HRRR-Smoke are defined following the IGBP land cover classification (17 land
cover types). The plume-rise in the model is based on Freitas et al. (2007) with heat energy flux
estimation parameterized as a function of FRP per unit fire size. HRRR-Smoke forecasts and
simulations have been comprehensively evaluated for several fire seasons. These evaluations have
included comparisons with hourly $PM_{2.5}$ measurements from the U.S. EPA Air Quality System
Network at multiple sites in the Washington state during the 2015 fire season (Deanes et al., 2016).
The HRRR-Smoke model forecasts for FIREX-AQ were evaluated by Ye et al. (2021) using
aircraft in-situ and remote sensing measurements.

## 320   3. Data

## 321   3.1. GOES-16 Fire Product

GOES-16/GOES-East was the first in NOAA's GOES-R series of geostationary satellites. It was
launched in November 2016 and occupies an orbit over 75.2°W. The **A**dvanced **B**aseline **I**mager
(ABI) is a 16-channel (2 visible, 4 near-infrared, 10 infrared) passive imaging radiometer onboard
GOES-16. It provides imagery of the Earth's surface and the atmosphere at very high spatial (2
km for infrared bands) and temporal (5 min for CONUS, 15 min for the Western Hemisphere/Full-
Disk) resolutions and includes several features that can be used to improve fire detection and
emissions estimation. For example, the finer spatial and temporal resolution of ABI data would
enable detection of small and short-lived fires. Under clear sky conditions, the minimum detectable
size of a fire (mean temperature: 800K) is estimated to be 0.004 $km^2$ at the sub-satellite point.



Short-lived fires are often missed by polar-orbiting satellites due to their limited temporal coverage.

The **F**ire **D**etection and **C**haracterization (FDC) product is one of the multiple GOES-16 ABI derived baseline products. The product has a spatial resolution of 2 km and is available for CONUS every 5 minutes. It uses a modified version of the Wildfire Automated Biomass Burning Algorithm (WF-ABBA) (Prins and Menzel, 1992;1994;Prins et al., 1998;2001;Schmidt and Prins, 2003) developed specifically for the ABI (referred to as ABI algorithm hereon). The ABI algorithm primarily relies on retrievals in the 3.9 and 11.2 µm spectral bands (ABI channels 7 and 14) and channel 2 (if available during daytime) to identify fires and derive sub-pixel fire properties in a two-step process consisting of identifying potential fires and subsequently filtering out false alarms. The algorithm uses several ABI (brightness temperatures/radiances (Channels 7 and 14 required, Channels 2 and 15 are optional), solar geometry and ABI sensor quality 3BEM flags) and non-ABI datasets (Global land cover classification, land/sea/desert mask from MODIS 5 collection, NCEP total precipitable water, MODIS global emissivity) in the process of deriving the final fire product. The product provides fire detection locations (latitude, longitude), fire properties (e.g., sub-pixel instantaneous fire size, fire radiative power, fire brightness temperature etc.) and a metadata mask classifying each detection into one of six categories (Code 10(30): Processed fire (sub-pixel fire size and temperature estimated), Code 11(31): Saturated fire pixel, Code 12(32): Cloud contaminated (partially cloudy/smoke), Code 13(33): High probability fire, Code 14 (34): Medium probability fire and Code 15(35): Low probability fire. The codes in parenthesis are used when the detection also passes a temporal filtering test). We only use Codes 10(30) in this study due to the availability of both FRP and fire size estimates. The sub-pixel instantaneous fire size and temperature is estimated using the Dozier technique (Dozier, 1981).





The Dozier method utilizes the total radiances in the 3.9 and 11.2 µm spectral bands and the
respective radiances in these bands from the fire and the background to solve for the proportion of
each ABI pixel that is on fire. Under realistic conditions (likely to be encountered in an operational
environment), Giglio and Kendall (2001) estimated that the random errors (at one standard
deviation) in estimating the fire size could be within 50% when the proportion of the pixel on fire
is more than 0.005. For proportions lower than 0.005, both the systematic and random errors could
be greater. GOES-16 data for the FIREX-AQ campaign period was available publically.

## 3.2. NASA DC-8 Airborne Observations from FIREX-AQ


### 3.2.1. Black Carbon Measurements from the NOAA Single-Particle Soot Photometer (SP2)

We use refractory Black Carbon (rBC) measurements from the NOAA Single Particle Soot
Photometer (SP2) (Schwarz et al., 2006;2008;2010a;2017;Perring et al., 2017) to evaluate WRF-
Chem simulated BC. Henceforth, we use the terminology BC to refer both to the material
quantified by the SP2, and the modeled species. The SP2 is primarily used to measure the
refractory Black Carbon (rBC) mass content of individual accumulation mode aerosol particles.
These mass estimates are independent of the particle mixing state or morphology. The instrument
has been used on various research aircrafts to provide airborne rBC in-situ measurements in
multiple field campaigns (e.g., NASA DC-8 (SEAC4RS) (Perring et al., 2017), NSF/NCAR GV
(HIPPO)(Schwarz et al., 2010b)). The SP2 flew onboard the NASA DC-8 for both the Boise and
Salina phases of the FIREX-AQ field campaign and provided in-situ measurements of rBC mass
concentration (ng -BC/std. m$^3$, (1013 mb pressure and 273K temperature) at 1-Hz frequency. The





rBC concentrations reported by the SP2 include final calibrations and adjustments for dilutions, a
correction factor to account for the non-detected rBC (sizes outside of SP2 detection range (90-
550 nm)) as well as rejection of highly contaminated (due to high concentrations) observations.
Smaller concentration biases also occurring under high aerosol loadings (Schwarz et al., under
review 2021) but affecting rBC concentrations by well less than 20% have not been corrected.
These biases are negligible in the context of the model comparison here.

**3.2.2. Organic Aerosol Measurements from the University of Colorado Boulder Aircraft**

**High-Resolution Time-of-Flight Aerosol Mass Spectrometer**
We use Organic Aerosol (OA) mass concentration measurements from The University of Colorado
Boulder Aircraft High-Resolution Time-of-Flight Aerosol Mass Spectrometer (CU HR-ToF-
AMS) and use the provided OA/OC ratio (based on (Aiken et al., 2008;Canagaratna et al., 2015))
to derive OC concentrations for comparison to the WRF-Chem simulated OC concentrations
(Note: OA/OC is not computed for OA values under the detection limit, and for those datapoints
a value of 1.8 OA/OC was used, consistent with the GOCART assumptions). The CU HR-ToF-
AMS (DeCarlo et al., 2006) can be used to perform high temporal resolution (demonstrated ability
of measurements at 0.1 Hz (Guo et al. (in prep)) measurements of bulk organic aerosol with
extensive characterization of its intensive properties (e.g., O/C, H/C, PMF factors) and inorganic
salts (e.g., ammonium sulfate ($(NH_4)_2SO_4$) , nitrate ($NH_4NO_3$) and chloride ($NH_4Cl$)) in submicron
(up to 900 nm vacuum aerodynamic diameter (Guo et al., 2021)). It is one of the several available
versions of the AMS that incorporates an improved high-resolution mass spectrometer. The
instrument takes in ambient air through a dedicated aerosol inlet (HIMIL (Stith et al., 2009)) into
an aerodynamic lens (residence time < 0.4 s) which directs the particles into a narrow beam. The





non-refractory particles are subsequently vaporized by impaction on a heated surface (600 °C) and
the vapors are ionized by electron ionization. Finally, these ions are analyzed by mass
spectrometry. The CU HR-ToF-AMS flew onboard the NASA DC-8 for both the Boise and Salina
phases of the FIREX-AQ field campaign. The instrument provided in-situ measurements at 1-Hz
frequency and switched to a higher time resolution of 5 Hz to sample fire plumes, especially the
smaller ones in the Salina phase.

**3.2.3. Aerosol Optical Property Measurements from the NASA Langley Airborne High**
**Spectral Resolution Lidar (HSRL)**
We use backscatter coefficient (532 nm) measurements from the NASA Langley airborne High
Spectral Resolution Lidar (HSRL) (Hair et al., 2008) to compare to WRF-Chem simulated
backscatter coefficient. WRF-Chem backscatter coefficient is computed using the ratio of the
WRF-Chem simulated aerosol extinction coefficient for different species (BC+OC, $SO_4^{2-}$, dust,
SS) and the corresponding lidar ratios. The lidar ratios are used from Burton et al. (2012). The
HSRL can provide measurements of aerosol backscatter and extinction coefficients (532 nm),
aerosol backscatter coefficient (1064 nm) and aerosol depolarization (532 nm and 1064 nm). The
instrument employs the HSRL technique at 532 nm and the standard backscatter lidar technique at
1064 nm. The HSRL technique relies on the differences in the spectral distributions of the
backscattered lidar signal from aerosols and molecules. The returned lidar signal is split into two
optical channels, namely the molecular backscatter channel and the total backscatter channel. The
molecular backscatter channel consists of an iodine ($I_2$) vapor absorption filter, which removes the
aerosol component of the returned lidar signal but passes the component due to molecules. The
total backscatter channel is non-selective and allows all frequencies to pass.





# 4. Results and Discussion

The Williams Flats wildfire began on August 2, 2019, 5 miles Southeast of Keller (Southwestern Ferry County) in Washington (WA) USA. The fire was caused by lightning strikes accompanying an early morning thunderstorm near the Colville Indian Reservation. The 100% containment date for the fire was reported to be August 25, 2019, and it burned an estimated 44,446 Acres (Source: Inciweb). The fire was the flagship fire of the Boise phase of the FIREX-AQ campaign and the focus of the DC8 science flights on August 3, 6, 7, and 8, 2019. These flights sampled both fresh and aged smoke plumes from the fire. On August 8, 2019, the fire also generated a pyro-cumulonimbus cloud (PyroCb) which was sampled by the DC8 science flight for the day. The Horsefly fire started on August 5, 2019, 15 miles east of Lincoln in the Lewis and Clark County (Montana) and burned 1274 acres in the first 24 hours. The fire was sampled on the flight of August 6[th]. The fire was reported to have burned 1350 acres till August 23, 2019 with zero growth reported in the prior week.

## 4.1. BC and OC Emission Estimates

Figure 3 shows the estimated BC and OC emissions (3BEM and FRP versions) for the Williams Flats fire on the DC-8 flight days (August 3- 8, 2019). Emissions from the Horsefly fire which was sampled on the August 6 flight are also shown. In general, the BC and OC emissions estimates from the FRP approach were significantly higher than the 3BEM approach on all flight days for the Williams Flats fire. For BC, the FRP-based emissions were 32 times higher on August 3, 2019 when Williams Flats was in its initial stages and varied between 12 to 47 times the emissions in the 3BEM version till August 8, 2019.



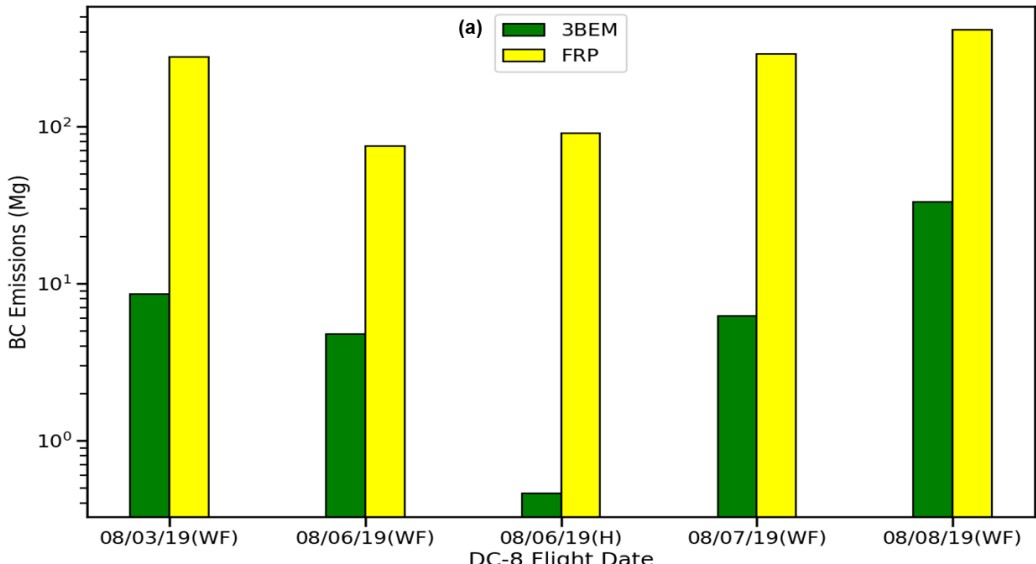

444

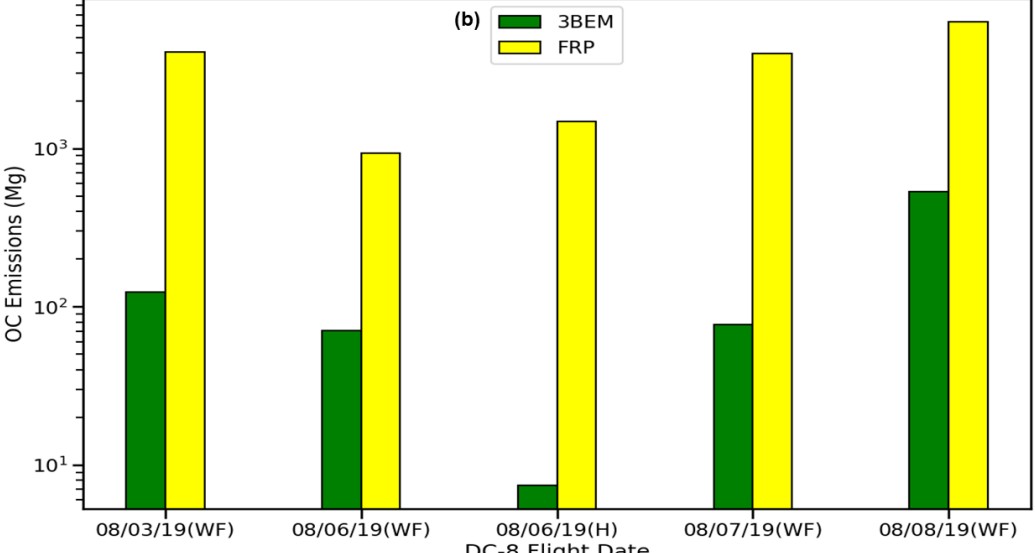

445

**Figure 3: Model-predicted BC (a, top) and OC (b, bottom) emissions from the Williams Flats (WF) fire on the DC-8 science flight days (August 3 – 8 2019) during FIREX-AQ. The emissions for Horsefly (H) fire on August 6, 2019, are also included (Bar set 3 for BC and OC).**



OC emissions also showed a similar trend with the FRP version emissions being 33 times higher
on August 3$^{rd}$ and 12-52 times higher for the remaining flight days. BC and OC emissions for both
approaches increased during August 3-8, 2019, with the maximum emissions observed on August
8, 2019 when Williams Flats generated a PyroCb event. The Williams Flats fire increased from
10,438 acres to 40,000 acres during August 3 -8, 2019 *(source: Inciweb August 4 and August 9,*
*9:00 am update)* which is reflected in the increase in BC and OC emissions. The increases were
much larger for the FRP based approach indicating that the FRP-based methodology is more
sensitive to the changes in fire behavior over time. Emissions in the 3BEM version were lower for
the Horsefly fire as well with the FRP based emissions being 198 times higher for BC and 200
times higher for OC. Thus, the FRP-based approach yields substantially higher emissions from
wildfires as compared to the 3BEM approach. The significant differences in emissions in the two
approaches could be attributed to the fundamental difference in the emissions estimation
methodology in the two approaches. The 3BEM approach uses the instantaneous fire size while
the HRRR-Smoke approach uses the FRP. Both these parameters could vary at substantially
different rates over the lifetime of a fire and therefore could lead to very different results. Ye et al.
(2021) compared the emissions between 12 different forecasting systems including WRF-Chem at
UW Madison (using GOES-15 fire product) and HRRR-Smoke and found that models using FRP-
based emission estimation approaches had substantially (mean factor of 5.6) higher emissions than
those using burned-area based (referred to as hotspot-based in their study) approaches.




## 4.2. Evaluation of WRF-Chem Simulations for DC-8 FIREX-AQ Science Flights (August 3 – 7, 2019)

This section includes a discussion of the relevant FIREX-AQ flights, interpretation of the FIREX-AQ aerosol observations and evaluation of the WRF-Chem model (3BEM and FRP versions) using FIREX-AQ observations of BC and OC, backscatter and also compares simulated plume heights with observed plume heights from the HSRL data. Plume height estimates are computed using the HSRL backscatter measurements and WRF-Chem simulated backscatter. Plume height is defined as the height at which the maximum change in the magnitude of the backscatter gradient is observed. We only focus on FIREX-AQ DC-8 science flights during August 3-7, 2019. We do not include the flight on August 8, 2019 in the analysis since the primary focus of this flight was on the Pyro-Cb produced by Williams Flats and current computational models do not have the capability to simulate these events. The WRF-Chem plumerise (in both 3BEM and FRP version) is a 1-D cloud model with a simplified microphysics scheme without any coupling between heat fluxes generated from fires and meteorology. Therefore, simulation of PyroCb events is beyond the capability of current computational models. Ye et al. 2021 also reported the current inability of models to represent the simulate PyroCb events based on their analyses of multiple forecasting models. However, recent work has focused on conceptual models that describe PyroCb (e.g., Peterson et al. (2017)) development during wildfire events. These models could serve as a starting point towards incorporating PyroCb simulation capabilities in current computational models.

For each FIREX-AQ DC-8 science flight, we first provide an overview of the flight followed by a qualitative comparison of the observations with WRF-Chem using HSRL flight curtains and finally quantitative comparisons between FIREX-AQ observations and WRF-Chem are discussed.



All altitudes reported are with respect to mean sea level (msl). We use the aircraft pressure altitude
to represent the aircraft altitude. The WRF-Chem Planetary Boundary Layer (PBL) height was
converted to the msl reference by adding the surface height to the WRF-Chem PBL variable.
**4.2.1 August 3, 2019, Flight**

The FIREX-AQ DC-8 science flight on August 3, 2019, involved extensive sampling of the
Williams Flats fire and a high altitude remnant of smoke associated with long-range transport. The
overall flight could be divided into two phases. Phase 1 was carried out at altitudes ranging from
2.7 – 3 km and sampling of the smoke plume extending 120 km downwind of the fire in the
northeast direction. Phase 2 extended 65 km downwind of the fire, initially in the northeast
direction and later in the eastern direction. The altitudes of sampling ranged between 3-3.4 km.
Figure 4 shows the WRF-Chem simulated aerosol optical depth (AOD) (3BEM (a, b) and FRP (c,
d) versions) for the Williams Flats fire at 00Z and 04Z with the DC-8 flight track overlaid. 00Z
represents the phase 1 of sampling while 04Z includes phase 2 and return to Boise. The 3BEM
experiment (Figure 4 (a, b)) simulated minor AOD enhancements (~ 0.3-0.6) due to the Williams
Flats fire. AOD enhancements were higher in the vicinity of the fire during phase 1 of sampling
(Figure 4(a), 00Z)) but dissipated later (Figure 4(b), 04Z, AOD: 0-0.2). In contrast, the WRF-
Chem FRP version simulated substantially higher AOD enhancements both near the fire as well
as in the transported plume downwind. These enhancements persisted throughout the DC-8
sampling period.


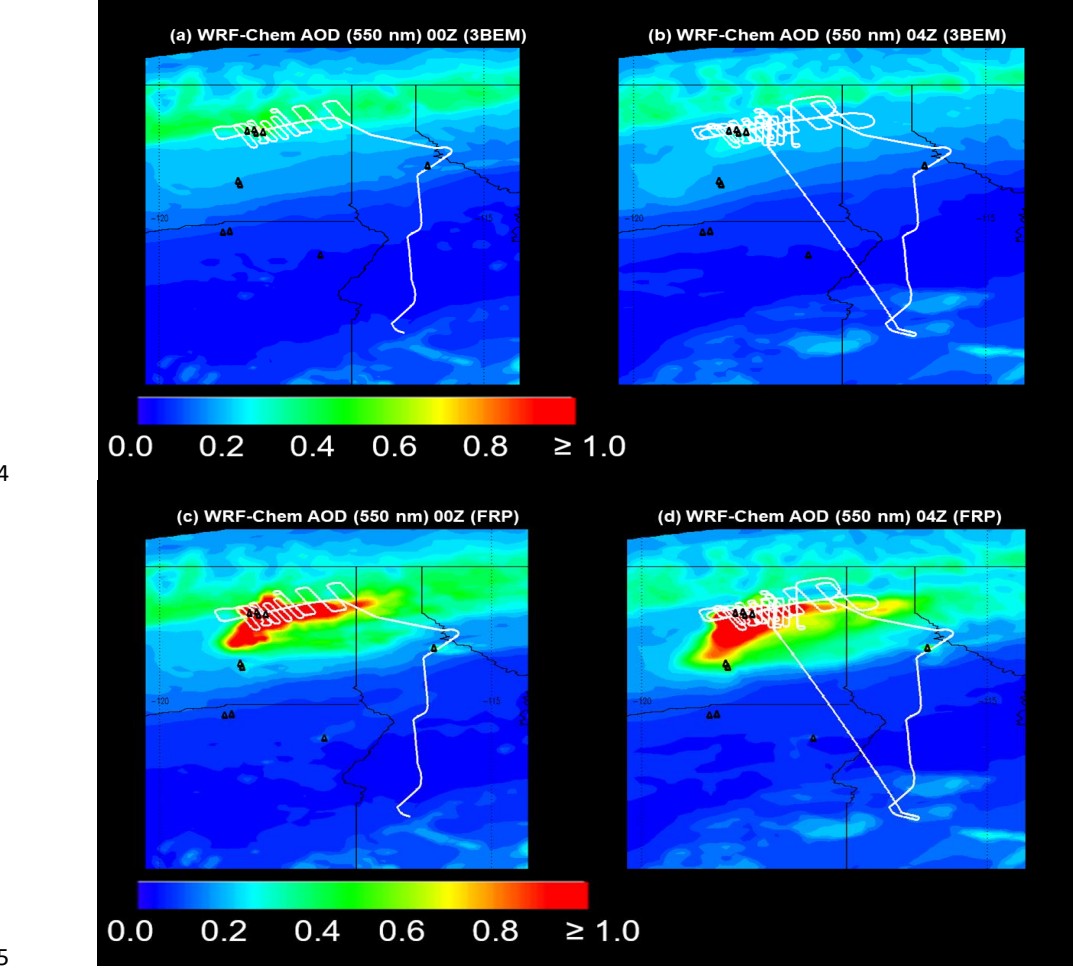


**Figure 4: WRF-Chem simulated aerosol optical depth (AOD) for the 3BEM (00Z (a, top left), 04Z (b, top right)) and FRP (00Z (c, bottom left), 04Z (d, bottom right)) versions during the FIREX-AQ DC-8 science flight on August 3, 2019. The DC-8 flight track is overlaid. The triangle markers indicate the locations of active fires.**

520

The lower AOD simulated by the 3BEM version is primarily due to the lower emissions (Section 4.1) in comparison to the FRP version while the decline in AOD during phase 2 could be due to the imposed diurnal cycle on emissions (maxima at 18Z) in this version. The 3BEM version simulated the plume formation and downwind transport of smoke towards the Northeast during



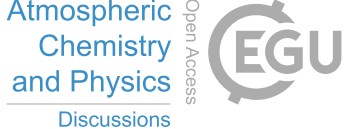

phase 1 but the decline in emissions in phase 2 resulted in a non-discernible plume with very low
AOD enhancements. In comparison, the FRP version simulated a far more intense plume with
AOD enhancements >=1 near the fire and in the east/southwest direction. The plume coincided
well with the sampling trajectory of the DC-8 indicating that the model simulated the spatial extent
of the plume reasonably well.
Figure 5 shows the curtains for HSRL aerosol backscatter coefficient (referred to as backscatter
hereon) measurements ((a)) and the simulated WRF-Chem backscatter (3BEM (b) and FRP (c)
versions). The DC-8 flight altitude is also shown. This science flight started with the DC-8 flying
over the Lick/Mica Creek fire on way from Boise to Williams Flats. The HSRL measurements
show the plume from the fire (~ between 21:00Z and 21:30Z) reaching an altitude of ~ 3 km. These
enhancements were underestimated by both the 3BEM and FRP versions possibly due to an
underestimation in emissions for this fire. The subsequent time periods in the HSRL observations
represent the DC-8 sampling phases of Williams Flats. Between 21:30Z and 22:00Z, the aircraft
travelled across Williams Flats to begin phase 1 of sampling. The phase 1 sampling period began
just after 22Z and continued downwind of the fire till 00Z followed by a return transit to the fire
(between 00Z and 1Z) and phase 2. The HSRL measurements show an alternating sequence of
high and low backscatter enhancements during phases 1 and 2 which represents the aircraft
traversing laterally in and out of the plume. The 3BEM version simulated localized backscatter
enhancements near the fire during the early stages of phase 1 (22Z – 23Z). These enhancements
were lower than the HSRL observations and declined significantly as the aircraft moved downwind
(23Z – 00Z) consistent with the observations. The enhancements in the downwind plume were
underestimated.









**Figure 5: FIREX-AQ DC-8 flight curtains for the August 3, 2019, science flight (a, top) HSRL observations, (b, middle) WRF-Chem 3BEM version and (c, bottom): WRF-Chem FRP version**

In phase 2, the 3BEM run simulated backscatter enhancements lower than that in phase 1 near the

fire (00Z – 01Z) which continued to decline as the aircraft moved downwind. The lower

enhancements in phase 2 as compared to phase 1 are consistent with the declining phase of the

emissions diurnal cycle in the 3BEM version. Thus, the 3BEM version showed several

discrepancies with the HSRL measurements which included underestimation of backscatter near

and downwind of the fire in both phases 1 and 2. The FRP version showed better overall agreement

with the HSRL measurements simulating comparable backscatter enhancements to the HSRL

measurements during most parts of phases 1 and 2. The FRP version was also able to better capture

the observed variation in the aerosol backscatter as the aircraft traversed in and out of the plume

although the coarse spatial resolution of the model (8 km x 8 km) acts as a limitation in exactly

simulating the observed variation from the center to the edge of the plume. In phases 1 and 2, the

model simulated continuously high aerosol backscatter near the fire which was also observed by

HSRL. It was also able to reproduce the variations in observed aerosol backscatter due to the

closely spaced legs of the DC-8 flight near the fire and widely spaced legs of the DC-8 flight

downwind of the fire in phase 1 (Figure 4(c)). For example, the alternate sequence of high/low

aerosol backscatter is wider for the widely spaced legs of the flight (downwind of the fire) as

compared to the closely spaced legs near the fire. The model was also able to reproduce the

variation in backscatter observed downwind of the fire very well especially in phase 1. Thus, the

model simulated a plume with high aerosol loadings near and extending a significant distance from

the fire which was more consistent with the observed plume as is evident in the better agreement

with the HSRL measurements. The FRP version appears to overestimate the plume height for

several parts of the flight (e.g., either side of 22Z, at 03Z, phase 1 and transit phase before phase



2) but showed better agreement with the HSRL measurements in the latter part of phase 2 (after
01Z) when the fire had intensified. Figure 6 (b-e) shows the time series of in-situ measurements
of BC (SP2)/OC (AMS) and the WRF-Chem simulated BC and OC (3BEM and FRP) along the
DC-8 flight track. The DC-8 altitude along with the WRF-Chem PBL height are also shown (a).
The 3BEM version was up to a factor of 100 lower than the in-situ BC measurements in phase 1
of sampling and up to ~ 250 times lower in phase 2. For OC, the 3BEM version underestimated
the measurements by up to ~ 125 times in phase 1 and up to more than 300 times in phase 2. These
results are consistent with the low AOD, and backscatter simulated by this version and can be
attributed mainly to the low emissions. The greater underestimation in phase 2 for BC and OC
could be due to the diurnal cycle imposed on the emissions. The higher emissions in the FRP
version contribute to the substantial improvements in the simulated BC and OC concentrations
resulting in better agreement with the SP2 and AMS in-situ measurements throughout the flight
period. The FRP version was able to reproduce the BC and OC enhancements observed near the
fire and downwind well, with the simulated BC being up to a factor of ~ 91 higher than the 3BEM
version, while for OC, the FRP version was up to ~28 times higher. Thus, the FRP version showed
a significant reduction in discrepancies between WRF-Chem and the SP2/AMS in-situ
measurements.







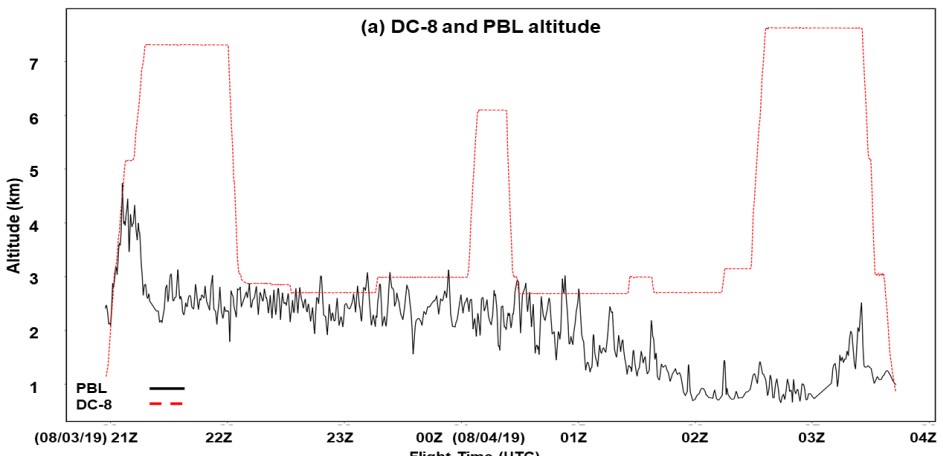







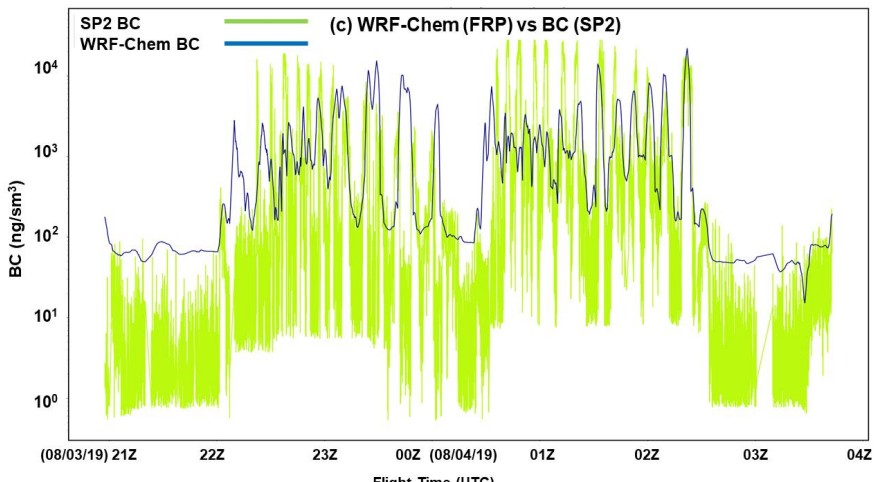


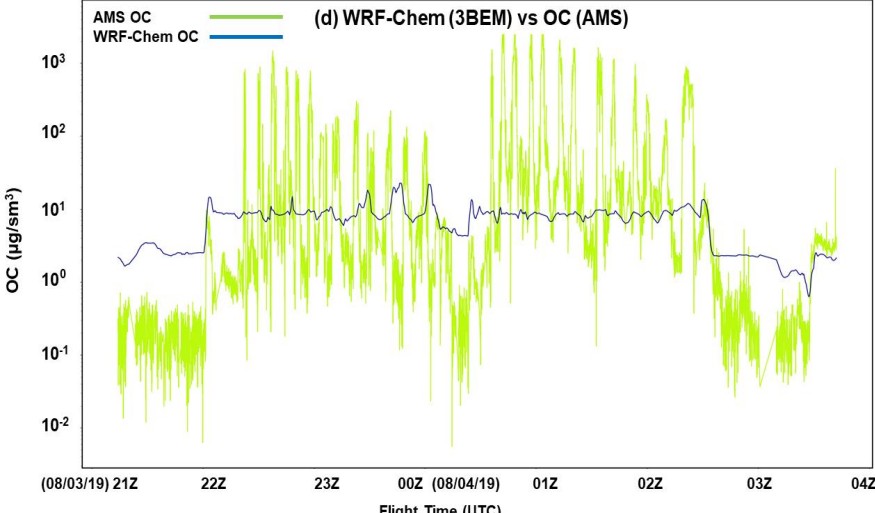






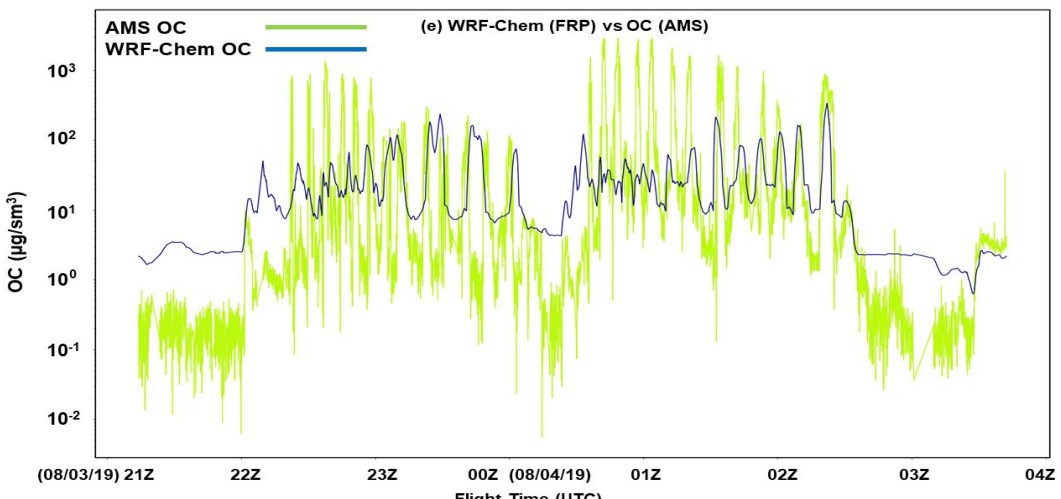


**Figure 6: (a) The DC-8 flight altitude (red) and the WRF-Chem planetary boundary layer height (black). Time series for BC (SP2) in-situ measurements and corresponding WRF-Chem simulated BC (3BEM (b) and FRP (c) versions), OC (AMS) in-situ measurements and corresponding simulated OC (WRF-Chem 3BEM (d) and FRP (e).**


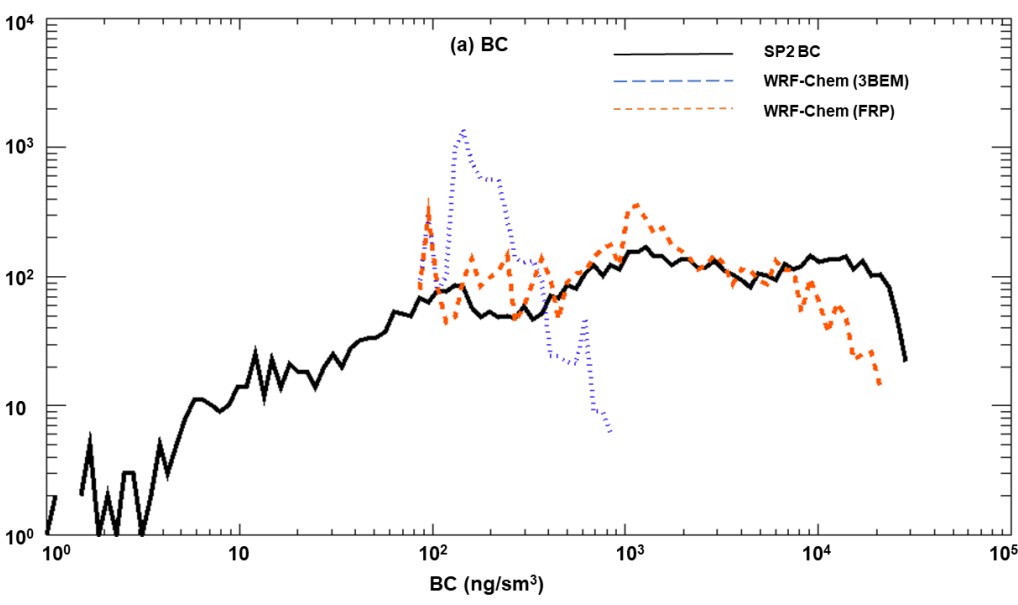










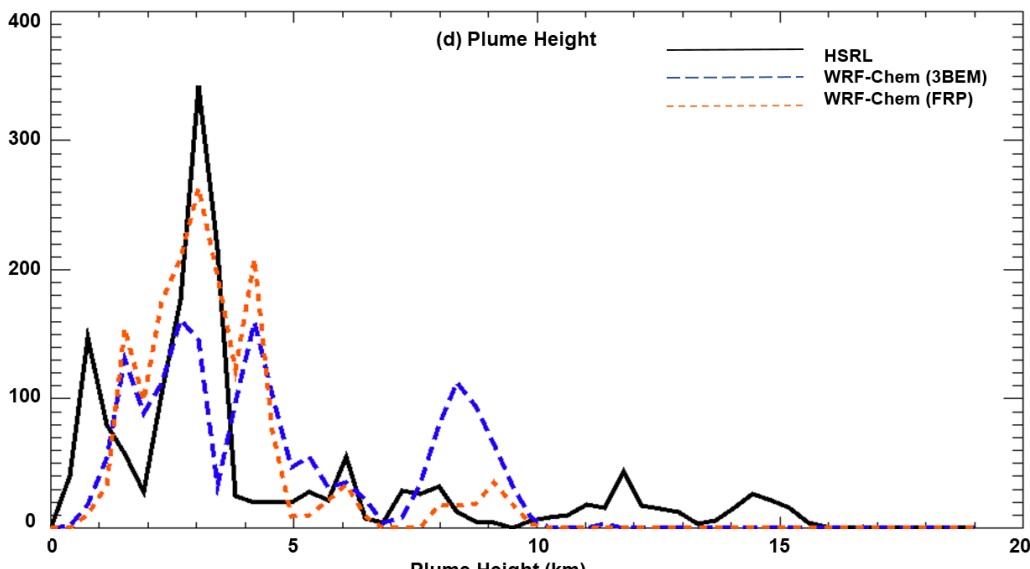


**Figure 7: Distribution functions for BC (a), OC (b), aerosol backscatter (c) and estimated**
**plume heights (d). Note: BC and OC only represent in-plume cases.**


Figure 7 shows the comparison of distribution functions of the in-situ measurements vs WRF-
Chem (3BEM and FRP runs) for BC and OC, backscatter, and the estimated plume heights. The
BC and OC distributions only account for the cases when the aircraft was in a smoke plume. The
backscatter and plume height distributions are based on all observations during the flight period.
For BC and OC, the in-situ measurements spanned a wide range (BC: 1 to $> 10^4$ ng/sm$^3$ and OC:
~0.1 to ~3000 µg/sm$^3$) which reflects the contrasting aerosol concentrations in the environments
in which the aircraft sampling occurred. For example, sampling included the center/edges of the
Williams Flats plume both near and a significant distance downwind from the fire as well as
remnants of any pollution at high altitudes. Aerosol concentrations in both cases could be very
different considering that the flight sampled fresh Williams Flats smoke while the pollution
remnants at high altitudes would have undergone significant dilution and thus would have much
lower aerosol concentrations. WRF-Chem (3BEM and FRP versions) showed less variability in





the simulated BC and OC concentrations than the measurements which could be due to the coarse
spatial resolution of the model and simplified chemical mechanism in the GOCART scheme. The
3BEM version captured very little of the observed variability in the BC and OC measurements
distributions. It simulated BC concentrations most frequently between ~80-250 ng/sm$^3$ and OC
concentrations between ~ 4-10 µg/sm$^3$ with a small fraction of higher values (BC: 250-900 ng/m$^3$,
OC: 10-11 µg/sm$^3$). The FRP version had an identical distribution for the lower end of
concentrations (BC: 80-100 ng/sm$^3$, OC: 4-6 µg/sm$^3$) which is representative of the remote
atmosphere and high altitudes where the impacts of changes in emissions and the plumerise are
negligible. The FRP version was able to reproduce the observed distribution to a much better
extent, especially for the high BC and OC concentrations (BC > 105 ng/sm$^3$), OC > 80 µg/sm$^3$)
relevant for large wildfire events, reflecting the improvements due to higher emissions. The high
biases in both versions of the model for the frequency of lower end concentrations (BC < 80
ng/sm$^3$, OC < 3 µg/sm$^3$) could correspond to the cases when the DC-8 was at the plume-edge or
when environments with low aerosol concentrations were being sampled (e.g., the long-range
transport plume). The model with its coarse spatial resolution (8km x 8km) could not accurately
simulate the variability observed while transiting from the center of the plume to the edges.
The backscatter distributions were similar to the BC and OC distributions except that the model
was closer to the measurements even though it was underestimating BC and OC as shown in Figure
7 (a, b). A potential reason for this discrepancy could be that we use lidar ratios from previous
work in deriving the backscatter from the WRF-Chem aerosol extinction coefficient. In addition,
meteorological parameters (e.g., relative humidity) and multiple aerosol species properties are
used in computation of aerosol optical properties which could result in biases in the estimation.
The backscatter distributions were identical for the 3BEM and FRP versions for low values (< 0.7


$Mm^{-1}Sr^{-1}$). These values could represent the high altitude phases of the flight during transition
from Boise to Williams Flats where the effects due to fires would not be a factor. Similar to the
BC and OC distributions, the FRP version captured the observed backscatter distribution well
especially for the higher values which were due to Williams Flats. The best estimated plume
heights based on HSRL observations were ~ 3 km (represented by the highest peak in Figure 7(d))
during the flight. In contrast, both 3BEM and FRP versions showed additional peaks in their
distribution functions on either side of the observed peak. Therefore, the predicted plume heights
varied between 2.7 – 4.1 km for the 3BEM version and 3 – 4.1 km for the FRP version. The FRP
version did produce a better agreement with the observed plume heights based on the highest peak
in the distribution function but also overestimated the heights for some parts of the flight.
Moreover, the low elevation smoke (represented by the peak < 1 km in HSRL) was either not
captured or overestimated (peak ~ 1.5 km) by both WRF-Chem versions.
**4.2.2. August 6, 2019 Flight**
The FIREX-AQ DC-8 science flight for August 6, 2019, had two targets namely, Williams Flats
and the Horsefly fire in Montana. Williams Flats was sampled first followed by an extensive
sampling of Horsefly which spanned more than 200 km downwind of the fire. For Williams Flats,
the sampling could be divided into two phases with phase 1 focusing on sampling low elevation
smoke and phase 2 involving sampling of the fire plume at a higher altitude (~3 km). Figure 8
shows the WRF-Chem simulated AOD at 22Z and 00Z respectively (3BEM (a, b)) and FRP ((c,
d)) versions) on August 6, 2019, during different stages of the DC-8 science flight. At 22Z, Figure
8 (a) shows the Williams Flats sampling (phase 1 and 2) whereas at 00Z (Figure 8(b)), the Horsefly
sampling is included as well.

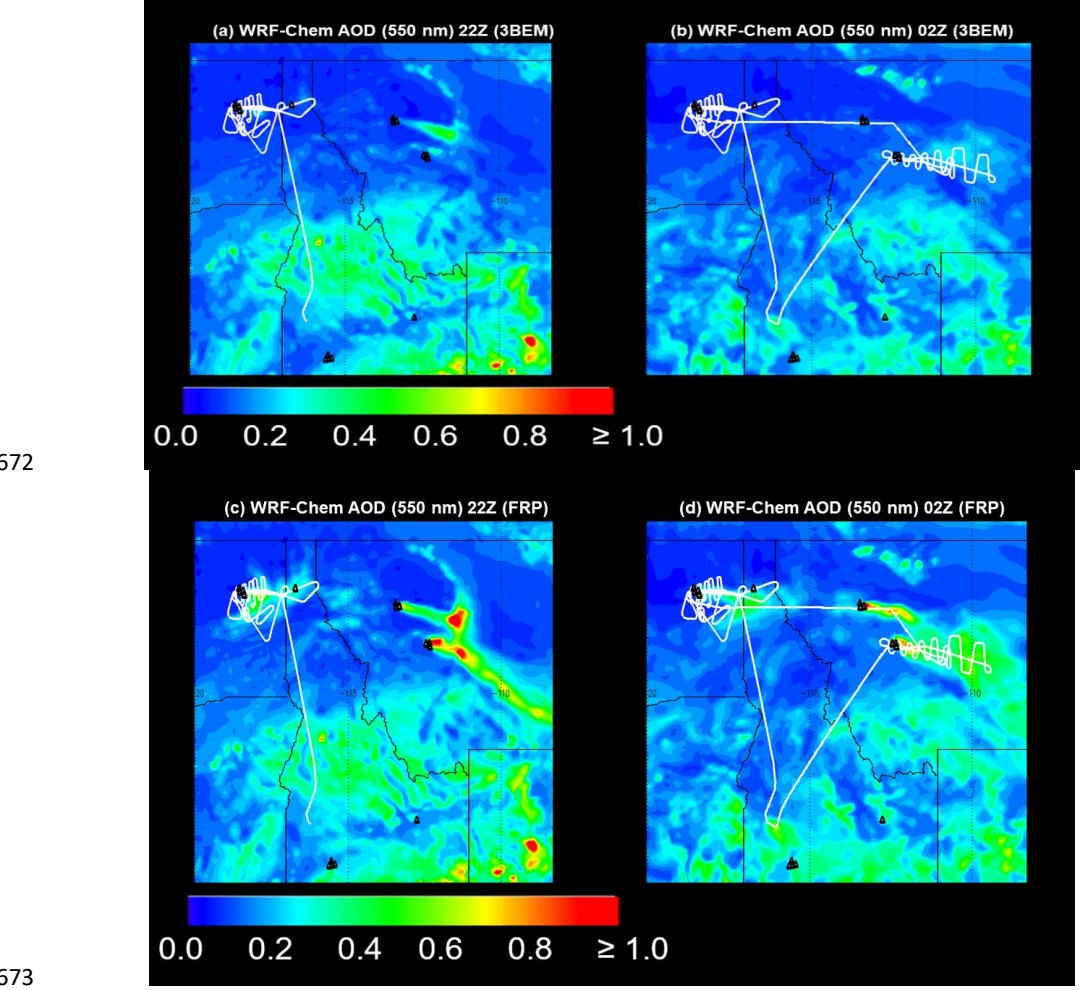



**Figure 8: WRF-Chem simulated aerosol optical depth (AOD) for the 3BEM (22Z (a, top left), 02Z (b, top right)) and FRP (c, 22Z (bottom left), 02Z (d, bottom right)) versions during the FIREX-AQ DC-8 science flight on August 6, 2019. The DC-8 flight track is overlaid. The triangle markers indicate the locations of active fires.**


The simulated AOD enhancements in the 3BEM version (0.0 – 0.3) were again lower than those
in the FRP based experiments with either thin/no noticeable plumes from both Williams Flats and
Horsefly over the flight period. On the other hand, the FRP version simulated well defined plumes
with higher AOD (0.3- >=1.0) for both fires and the spatial location and extent of the plume were





in good agreement with the DC-8 sampling legs. The Horsefly fire plume is represented very well
by this version (Figure 8(c, d)) based on the DC-8 sampling pattern. Similar agreement was
observed for the plume from Williams Flats which was predominantly towards the East. The
estimated emissions for Williams Flats were lower for August 6 as compared to the other flight
days, which resulted in the relatively lower AOD enhancements simulated than those on August 3

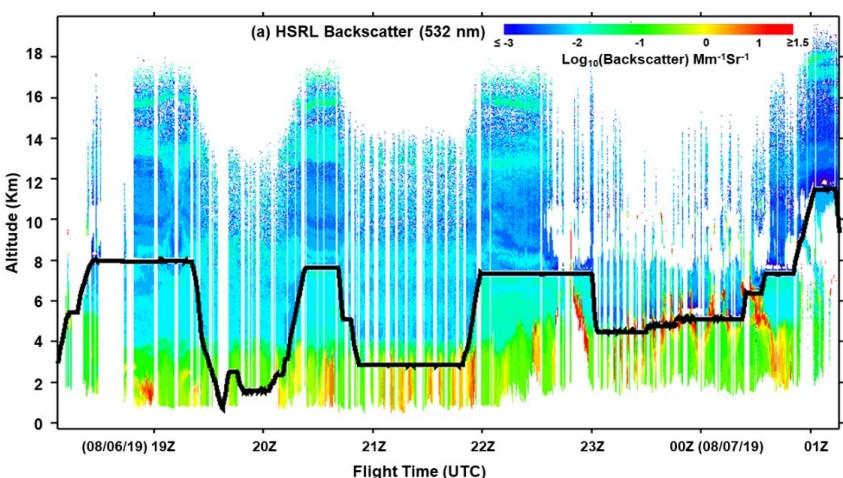


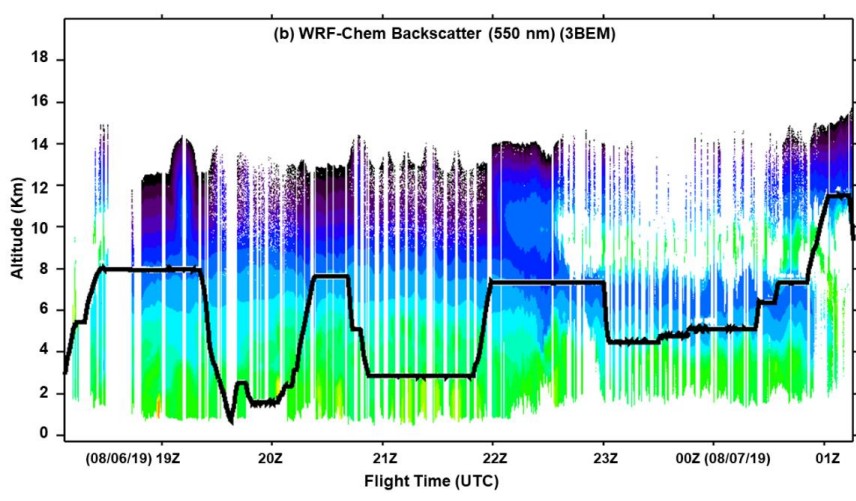


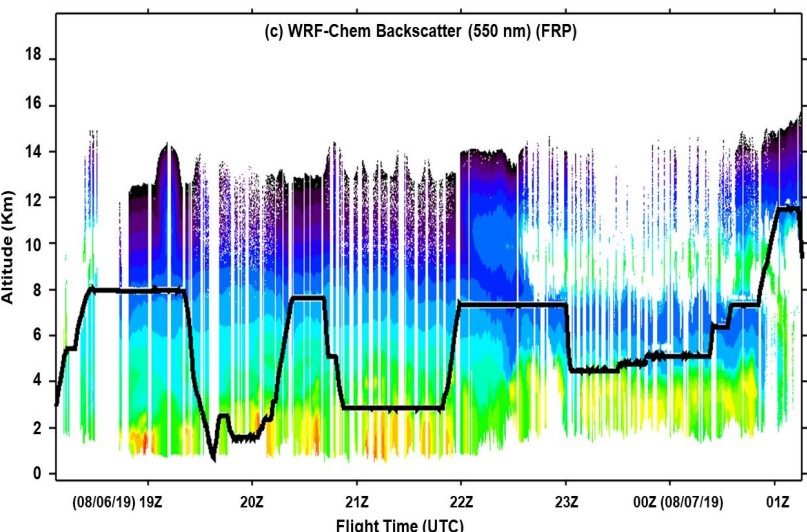


**Figure 9: FIREX-AQ DC-8 flight curtains for the August 6, 2019, science flight (a, top) HSRL observations, (b, middle) WRF-Chem 3BEM version and (c, bottom): WRF-Chem FRP version**


Figure 9 shows the curtains for HSRL backscatter measurements ((a)) and the simulated WRF-Chem backscatter (3BEM (b) and FRP (c) versions). The DC-8 flight track is also shown. The curtain represents the DC-8 sampling of the Williams Flats fire phase 1(between ~ 19:30Z and 20Z) and phase 2 (21Z to 22Z) and the Horsefly fire from 23Z to just before 00:30 Z. The backscatter enhancements during phase 1 (low level smoke sampling) were underestimated by the WRF-Chem 3BEM version while the FRP version tended to overestimate. The HSRL measurements were not available near 20Z (below the DC-8) due to attenuation which precludes any further comparisons. During 20Z-21Z, the high backscatter in the HSRL measurements correspond to Williams Flats as the DC-8 flew over the fire to begin phase 2 of sampling. These enhancements were largely absent in the 3BEM version but were reproduced well in the FRP version. During phase 2 of sampling (21Z-22Z), the 3BEM experiment only simulated sporadic backscatter enhancements which were biased low as compared to the HSRL measurements. The





measurements showed consistently high backscatter as the DC-8 traversed along the plume with
the alternating bands of high/low backscatter again reflecting the periods the aircraft was within
the plume or entering/leaving it. The FRP version did a better job than the 3BEM version,
simulating comparable backscatter enhancements to the HSRL measurements and represented the
variation along the flight track well. Between 22Z-23Z, the DC-8 travelled from Williams Flats
towards Montana to sample the Horsefly fire and flew over the Snow Creek fire and Horsefly
before beginning the sampling. The HSRL backscatter enhancements during this period were due
to these two fires and were better represented by the FRP version. For the Horsefly fire, the DC-8
travelled downwind in the plume starting at ~23Z and continuing sometime after 00Z, which was
followed by an upwind pass. The 3BEM version was biased low for this entire period consistent
with the low emissions. The FRP version did simulate higher backscatter enhancements than the
3BEM version throughout this period, but it was unable to reproduce the peak enhancements in
the HSRL measurements. In addition, WRF-Chem (3BEM and FRP) underestimated the plume
height for Horsefly (<= 4 km) as compared to the HSRL observations (~ 4 - 6 km). Consequently,
the variation of the backscatter enhancements along the flight track does not agree with the HSRL
observations.




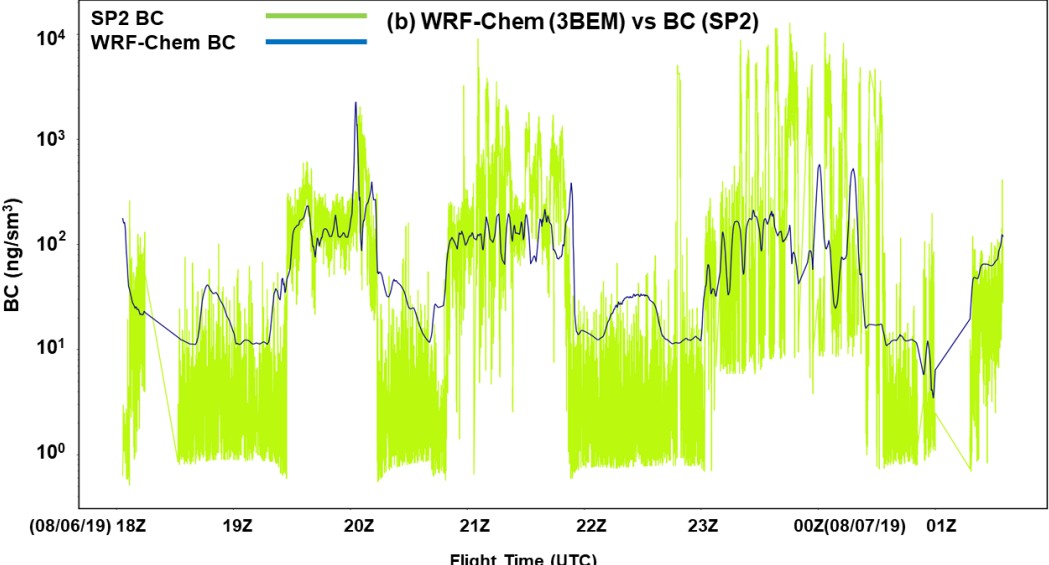






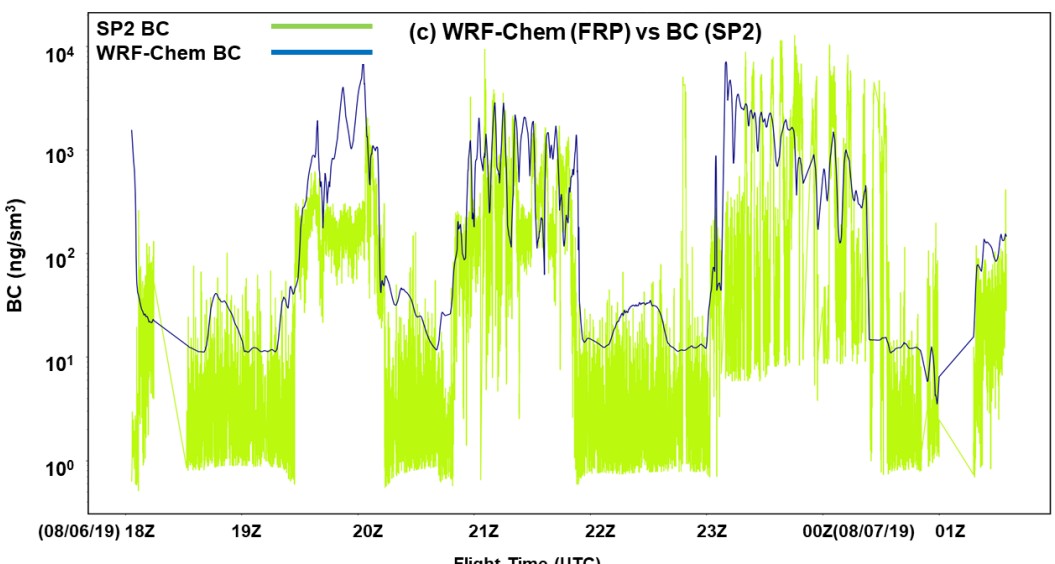


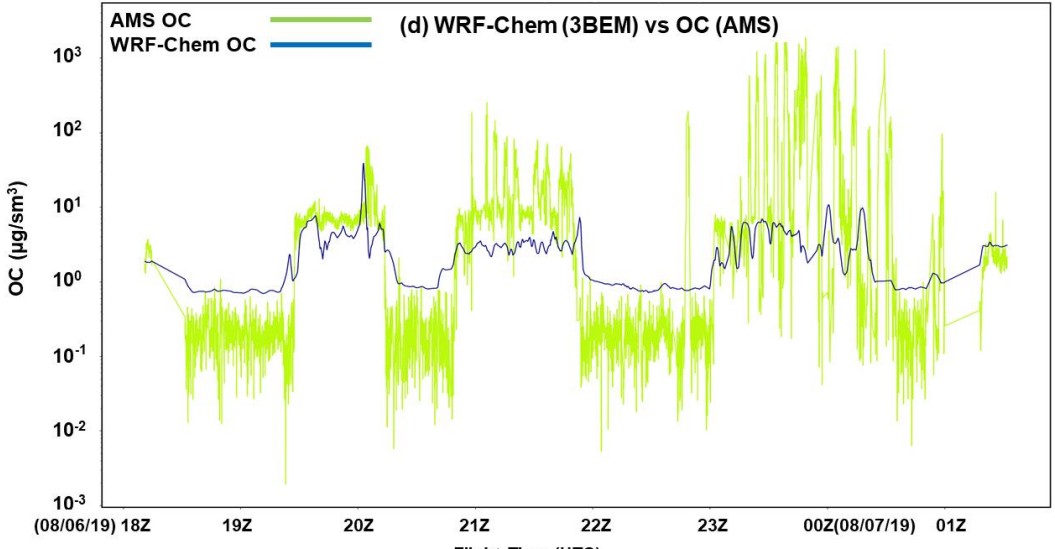









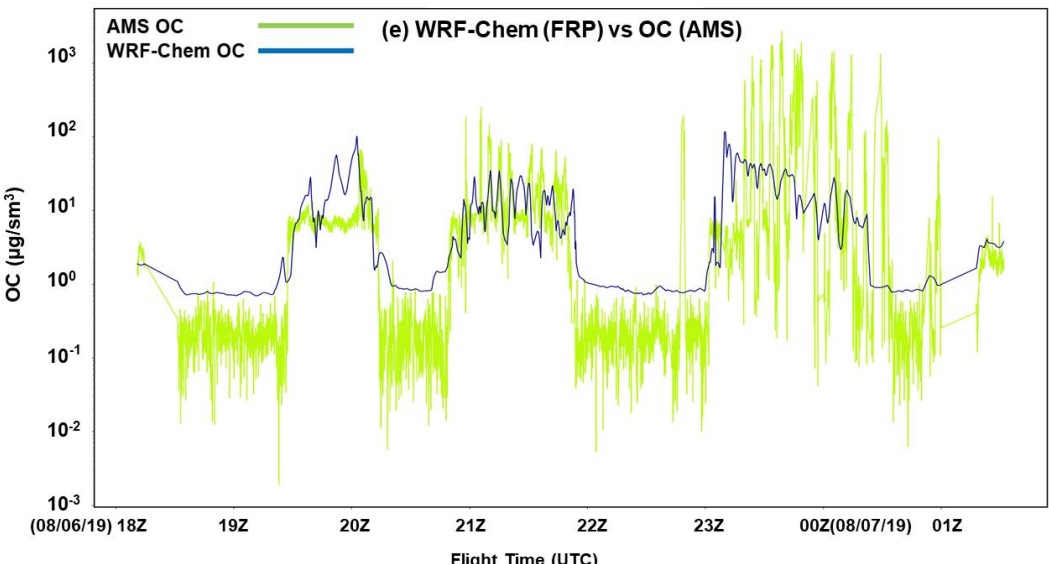


**Figure 10: (a) The DC-8 flight altitude (red) and the WRF-Chem planetary boundary layer**
**height (black). Time series for BC (SP2) in-situ measurements and corresponding simulated**
**BC (WRF-Chem 3BEM (b) and FRP (c) versions), OC (AMS) in-situ measurements and**
**corresponding simulated OC (WRF-Chem 3BEM (d) and FRP (e)).**

Figure 10 shows the time series of in-situ measurements of BC (SP2)/OC (AMS) and the WRF-

Chem simulated BC and OC (3BEM: 10 (a, c) and FRP: 10 (b, d)) along the DC-8 flight track.

The DC-8 altitude along with the WRF-Chem PBL height are also shown (a). The 3BEM version

was biased low for most part of the flight with the simulated BC up to 440 times lower than the

measurements and OC up to 1065 times lower. However, it performed better than the FRP version

in simulating the low elevation smoke as the FRP version significantly overestimated the BC and

OC concentrations (19Z – 20Z). The FRP version showed very good agreement for phase 2 of the

Williams Flats sampling, where it was able to simulate comparable concentrations of BC and OC

to the observations. For the Horsefly fire as well, the FRP version was able to simulate the high

BC levels observed but significantly underestimated OC. The FRP version simulated up to 125

times higher BC concentrations and up to 49 times higher OC concentrations than the 3BEM



version. The 3BEM version was biased very low for BC and OC during phase 2 of Williams Flats
and the Horsefly sampling. The BC and OC concentrations in the FRP version (Figure 10(b, d))
declined sharply as the DC-8 flew downwind of Horsefly, which could be attributed to an
underestimation of the injection heights or inability of the model to accurately simulate the
transport of the plume downwind resulting in lower plume heights than observed. The Horsefly
fire plume altitude increased downwind as shown in the HSRL backscatter measurements (Figure
9(a), 23Z onwards). This was accompanied by a gradual ascent of the DC-8 aircraft as it tracked
the fire plume (Figure 9 (a)). Since the plume-height was very low in the model, the BC and OC
concentrations along the flight track represented background level concentrations instead of the
enhanced levels caused by the fire. These concentrations declined even further as the aircraft
ascended in the later stages, which is observed in the time-series during the Horsefly downwind
sampling phase.

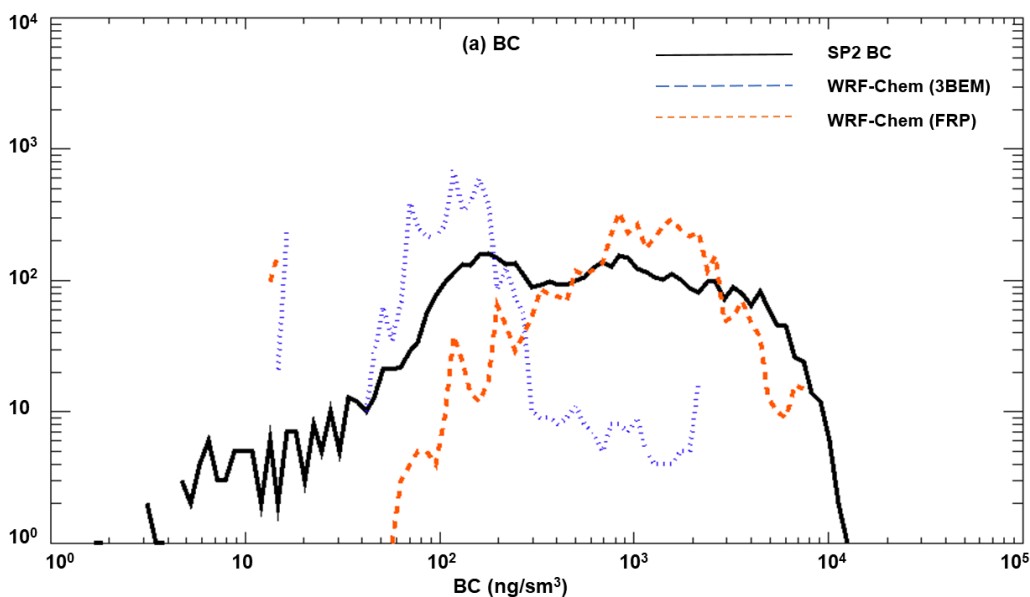










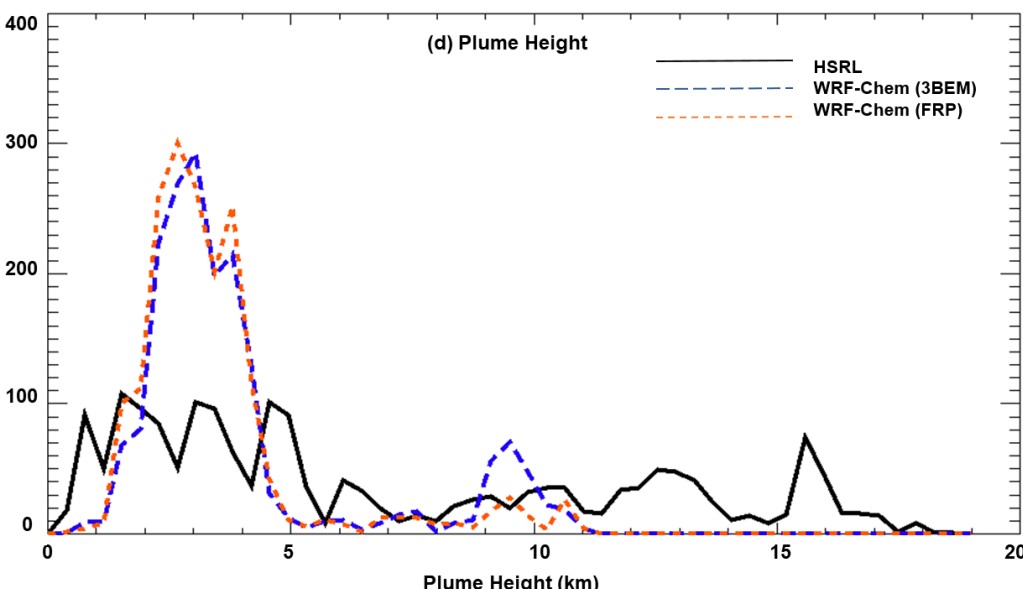


**Figure 11: Distribution functions for BC (a, top-left), OC (b, top-right), backscatter coefficient (c, bottom-left) and estimated plume heights (d, bottom- right). Note: BC and OC only represent in-plume cases.**

Figure 11 shows the comparison of the distributions of the measurements vs WRF-Chem (3BEM
and FRP runs) for BC, OC, backscatter, and the estimated plume heights. The BC and OC
distributions represent the in-plume data only. The observed distributions for BC and OC
represented a similar range of in-plume concentrations as the August 3rd flight, however, the lower
end of concentrations were higher for BC and OC, possibly due to this flight focusing only on
fresh smoke sampling unlike the August 3rd flight which also sampled aged smoke (long-range
transport plume). The significant variance of the BC and OC distribution also reflects the various
sampling conditions such as the aircraft traversing through the plume encountering high
concentrations at the center and lower concentrations towards the edges, the different altitudes of
sampling (phase 1 at lower altitude and phase 2 at higher altitude for Williams Flats) and traversing
downwind from the Williams Flats and Horsefly fires. Similar to the August 3 flight, the WRF-
Chem BC and OC distributions could not capture all the variability in the observations and were





also biased high primarily due to the coarse model resolution, which precluded accurate simulation
of the observed variability from the plume center to the edges. The 3BEM version distribution was
able to better capture the variability in the BC and OC distributions than for the August 3rd flight,
which was mainly due to the better simulation of BC and OC concentrations in the low-altitude
Williams Flats smoke. However, it still had a low bias compared to BC and OC measurements.
The FRP version showed good agreement with the BC distribution although it was biased low for
OC. The low bias could primarily be attributed to the underestimation during the Horsefly
sampling phase and the simplified chemistry in the GOCART mechanism (no SOA). Nevertheless,
the distributions for the FRP version showed both an increase in variability and a shift towards
higher simulated BC and OC concentrations. This resulted in better simulation of the variability in
the BC and OC measurements distribution as compared to the 3BEM version and improvements
in agreement with the observed BC and OC distributions at concentration levels relevant for fire
plumes. The backscatter distribution derived from the HSRL measurements showed similar
characteristics with lower values (< 0.01) primarily representing very high altitudes with no
influence of fire emissions. This region was identically simulated by WRF-Chem (3BEM and
FRP) since the primary differences between the two versions (fire emissions and plume-rise) had
little/no effects at these altitudes. The backscatter distribution also exhibited considerable
variability (values spanned six orders of magnitude) which was consistent with the high variability
observed in the BC and OC distributions. The backscatter distribution for the FRP version also
showed a shift towards simulating higher enhancements than the 3BEM version and showing better
agreement with the HSRL distribution at backscatter levels relevant to major fire events. The
plume heights distribution based on HSRL measurements showed several peaks which could be
attributed to the multiple altitudes at which smoke was sampled during this flight. Based on the





observed peaks, the heights could have ranged from 0.75 km to 6 km. The heights between 3 – 6
km are associated with the high altitude Williams Flats plume and the Horsefly fire plume while
the < 3 km altitude are from the lower altitude Williams Flats smoke. Neither WRF-Chem versions
could capture this variability in the observed plume heights distribution and simulated smoke
heights of ~ 3km (peak 1) and ~ 3.8 km (peak 2) for the 3BEM version (~ 2.7 and ~ 3.8 km for the
FRP version). Thus, WRF-Chem underestimated the plume heights for this flight, which as
discussed earlier in this section, could be a possible reason for the sharp decline in the simulated
BC and OC concentrations as the DC-8 proceeded downwind of the Horsefly fire.

### 810  4.2.3. August 7, 2019

The August 7, 2019, FIREX-AQ DC-8 science flight focused exclusively on the Williams Flats
fire with a four phase sampling strategy. Phase 1 involved sampling aged (transport age: one day
old) smoke from the fire which was transported eastward to Montana. This smoke was sampled
both in the East and West directions travelling along the axis of the plume. The remaining phases
focused on fresh smoke from the fire with phase 2 involving sampling at low altitudes (~ 3.7  -
4.3 km) and phases 3 and 4 involved higher altitude (~ 4.9 km) sampling. Figure 12 shows the
WRF-Chem simulated AOD at 23Z and 04Z respectively (3BEM (a, b)) and FRP ((c, d))
versions) on August 7, 2019, during different stages of the DC-8 science flight. The aged smoke
plume in Montana does not appear as a distinct feature in the WRF-Chem AOD plots possibly
due to the low simulated aerosol concentrations. Similar to the previous flights, the low
emissions and the diurnal cycle contributed to the 3BEM version simulating very small AOD
enhancements (0.2 - 0.6) which were prominent only during the early stages of the flight and
further declined during the fresh smoke sampling phase.


**Figure 12: WRF-Chem simulated aerosol optical depth (AOD) for the 3BEM (23Z (a, top -
left), 02Z (b, top-right)) and FRP (23Z (c, bottom-left), 02Z (bottom-right)) versions during
the FIREX-AQ DC-8 science flight on August 7, 2019. The DC-8 flight track is overlaid.**

The plume from the Williams Flats fire was only evident during the early stages of the flight and
was characterized by very low aerosol loadings. In contrast, the FRP version simulated very high





AOD enhancements (>=1) near the fire both before and during the fresh smoke sampling phase.
The model simulated a well-defined and persistent plume throughout the DC-8 sampling period.
The simulated plume coincided well with the DC-8 flight path during the fresh smoke sampling
phases.

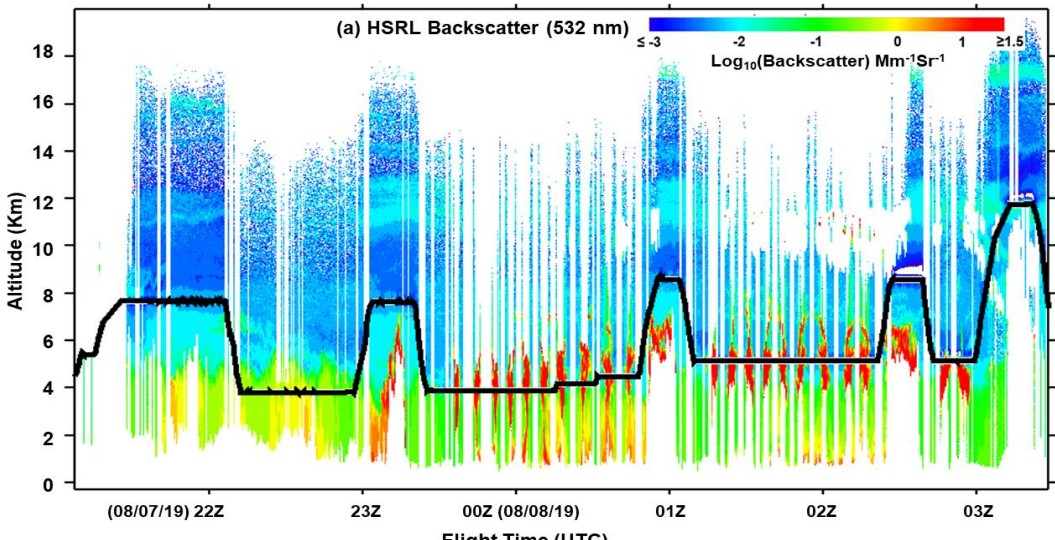


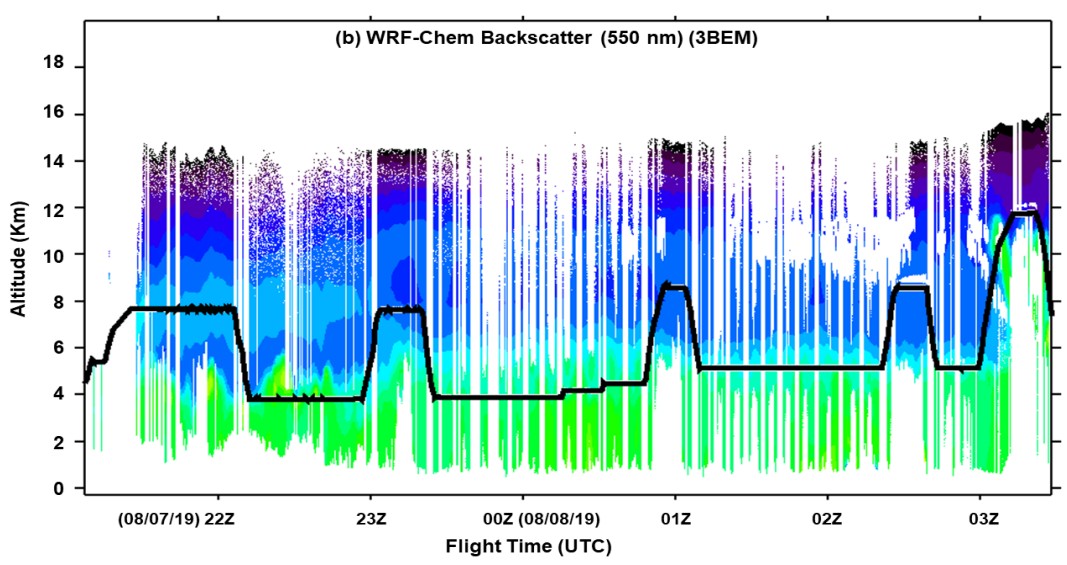


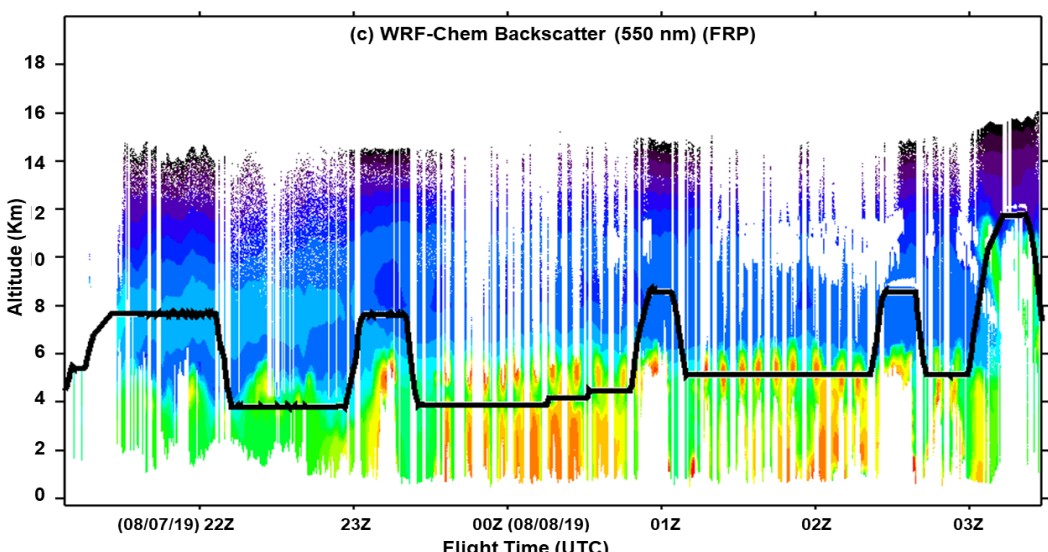


**Figure 13: FIREX-AQ DC-8 flight curtains for the August 7, 2019, science flight (a, top) HSRL observations, (b, middle) WRF-Chem 3BEM version and (c, bottom): WRF-Chem FRP version.**

Figure 13 shows the flight curtains for HSRL backscatter measurements and WRF-Chem
backscatter (3BEM and FRP runs) along with the DC-8 flight altitude. The flight sampled the aged
smoke from Williams Flats between 22-23Z at an altitude of ~ 3.7 km. The HSRL measurements
show the aerosol layer height to extend close to 6 km which was simulated very well by both the
3BEM and FRP runs although both versions were biased low. Subsequently, the DC-8 flew over
Williams Flats at an altitude close to 8 km to begin fresh smoke sampling and the HSRL
measurements showed very high aerosol backscatter during this period till ~ 7 km. This was
reproduced well by the WRF-Chem FRP version, however the altitude was underestimated (~ 5.5-
6 km) and for the 3BEM run, the backscatter enhancements were very low. During phase 2 of the
sampling as the DC-8 moved along the plume, the HSRL measurements showed high aerosol
backscatter values throughout with plume heights extending till ~ 6 km. The 3BEM version failed
to capture the observed enhancements and was biased low throughout the remainder of the flight





mainly due to the low emissions. The FRP version consistently simulated significantly higher
backscatter as compared to the 3BEM run and simulated the plume height between 5-6 km. Phase
2 was followed by a pass over the plume and phase 3 sampling. The observed plume heights during
this part of the flight ranged from ~ 5 – 6.5 km and the backscatter levels were high as shown in
the HSRL observations (01 – 02Z). The FRP version simulated enhancements comparable to the
HSRL observations but was still biased low. The vertical extents were ~ 5-5.5 km which were in
reasonable agreement with HSRL measurements. The backscatter observed during the last pass
over the fire at 8 km altitude was also well simulated by the FRP version with a plume height of ~
5.8 km matching well with that observed in the HSRL data (~ 6 km). During phase 4, the FRP
version showed significantly better agreements with the HSRL observations with higher
enhancements than the 3BEM run and a predicted plume height of ~ 5 km agreeing very well with
the HSRL observations (~ 5 km).

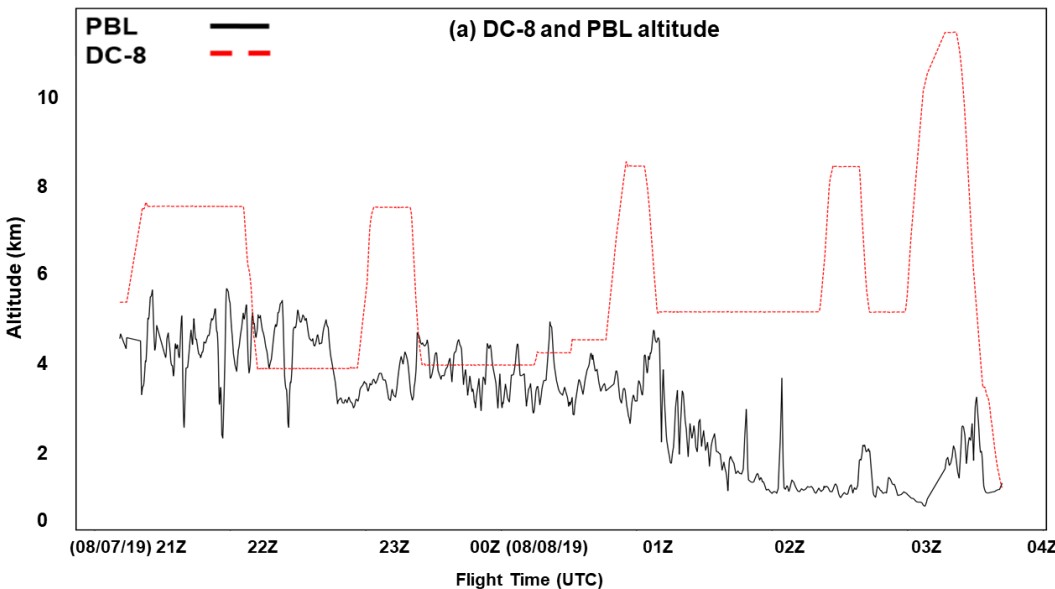








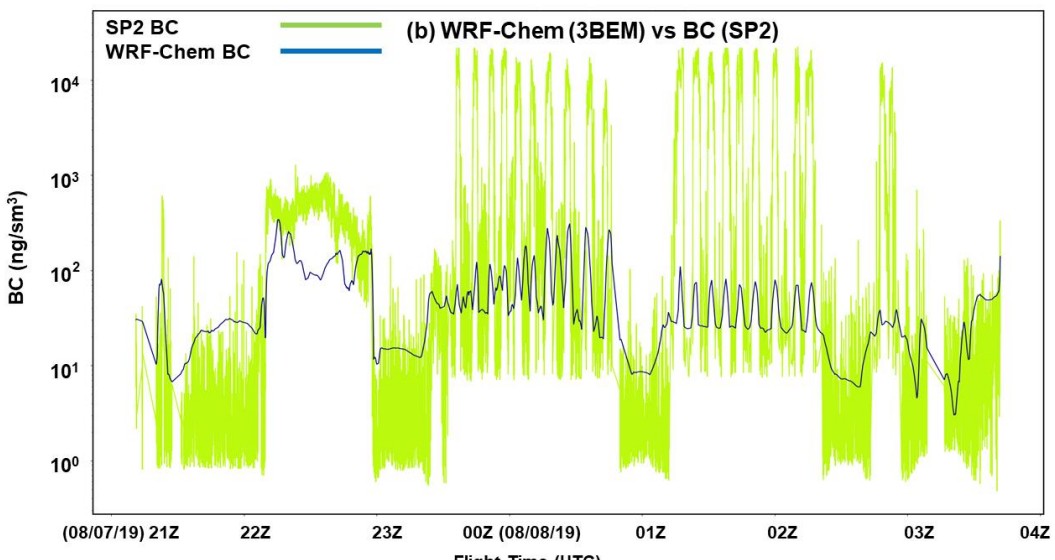








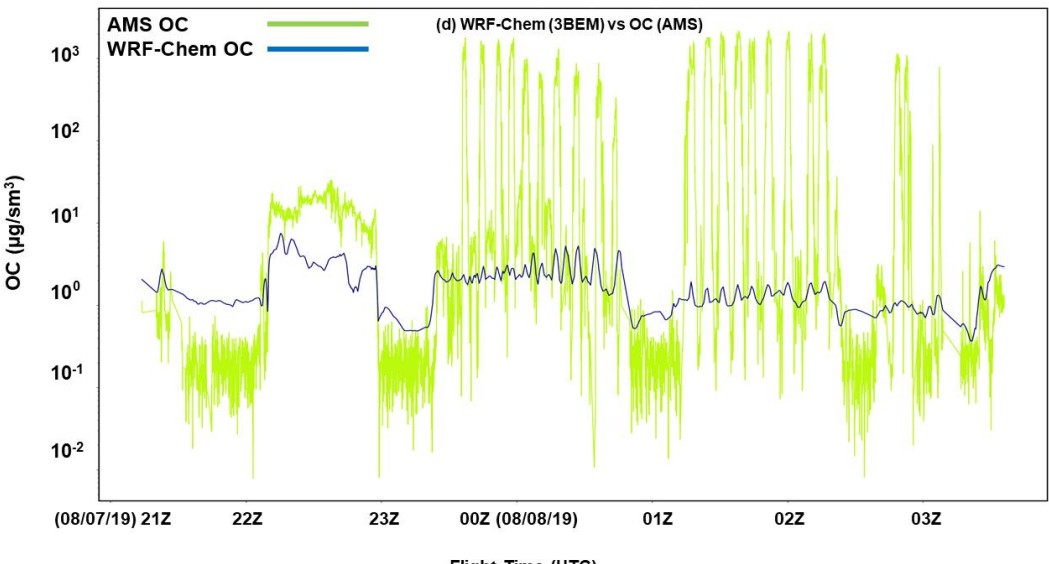


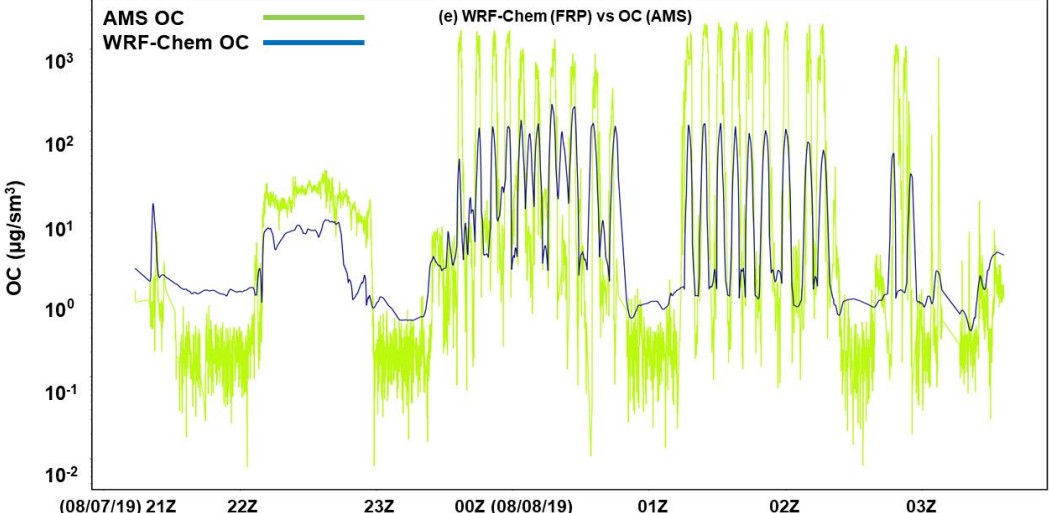


**Figure 14: (a) The DC-8 flight altitude (red) and the WRF-Chem planetary boundary layer height (black). Time series for BC (SP2) in-situ measurements and corresponding simulated BC (WRF-Chem 3BEM (b) and FRP (c) versions), OC (AMS) in-situ measurements and corresponding simulated OC (WRF-Chem 3BEM (d) and FRP (e)).**






Figure 14 shows the time series of in-situ measurements of BC (SP2)/OC (AMS) and the WRF-
Chem simulated BC and OC (3BEM: (b, d) and FRP: (c, e)). The 3BEM version was not able to
reproduce the observed BC and OC concentrations during any of the sampling phases. The
underestimations were up to 842 times for BC and up to 1439 times for OC.  The 3BEM version
performed particularly poorly in phases 3 and 4 of the flight where the low biases were very large
and could be caused by the low emissions in the later stages of the flight. In contrast, the FRP
version was able to reproduce the observations very well especially in the fresh smoke sampling
phases of the flight. The higher emissions in the FRP version resulted in BC concentrations up to
124 times higher and OC concentrations up to 78 times higher than the 3BEM version. Both the
3BEM and FRP versions underestimated the aged smoke which could be due to simplified
chemistry in the GOCART mechanism. The underestimation of OC in the model was larger than
BC which could also be a consequence of the simplified chemistry in the model.
Figure 15 shows the comparison of the distributions of the measurements vs WRF-Chem (3BEM
and FRP runs) for BC, OC, backscatter, and the estimated plume heights. The observed
distributions for BC and OC were similar to the previous flights, exhibiting high variability due to
the sampling of a wide range of aerosol loading environments. For example, the Williams Flats
aged plume was characterized by significantly lower aerosol concentrations as compared to the
fresh plume sampled later. In addition, similar to the previous flights, the concentrations at the
edge and center of the plume would also contribute to the variability observed in the BC and OC
observations distributions. WRF-Chem (FRP version) was able to reproduce a significant fraction
of this variability for BC and OC particularly for the high concentrations, as shown in
corresponding distributions. The backscatter distribution of the FRP version also showed better
agreement with the HSRL backscatter distribution. These major improvements, which were also



found in earlier flights, includes a significant shift in the BC and OC backscatter distributions
towards higher values and better agreement with observations. The estimated plume heights from
HSRL showed one prominent peak near 5 km which would correspond to the Williams Flats smoke
(aged and fresh). For the WRF-Chem 3BEM version, the simulated plume height varied between
3.5~ 5 km (based on the two peaks in the distribution), while the FRP version varied from 3.5 –
5.5 km. Thus, both versions showed significant variability in the plume heights which could be
due to different simulated injection heights in the model.

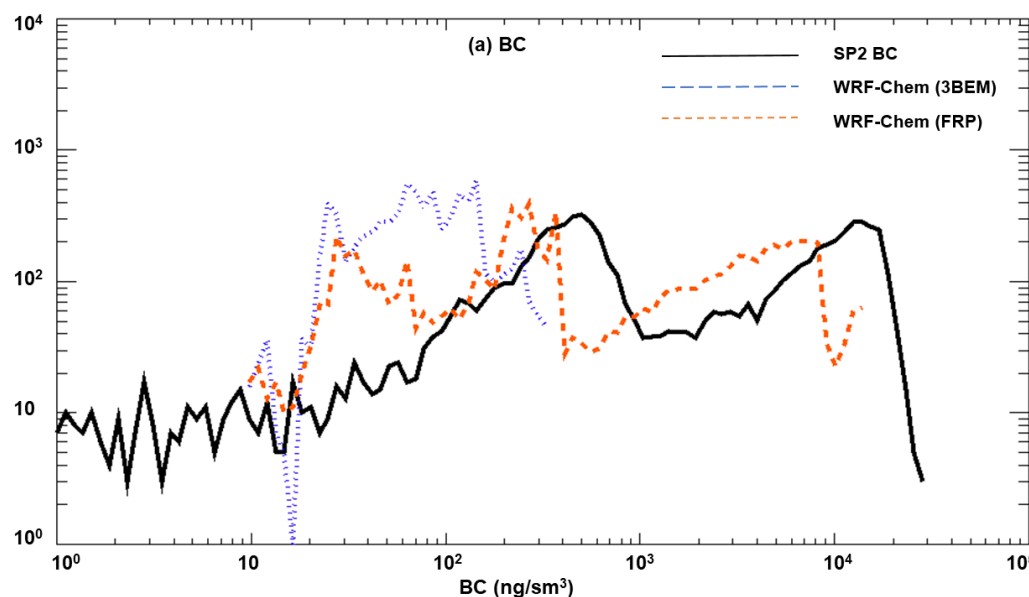




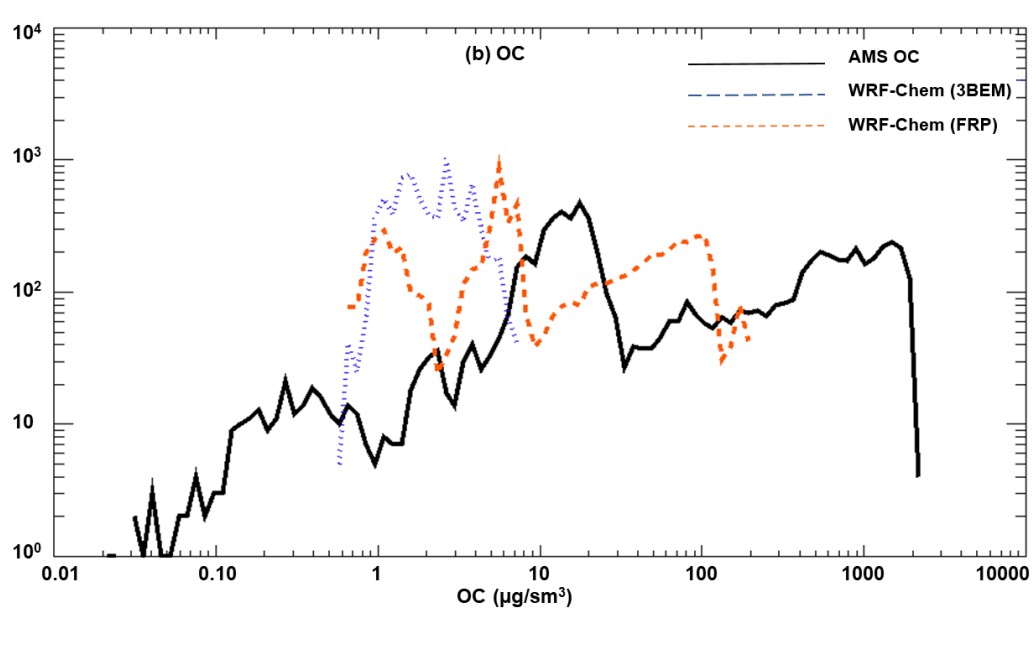


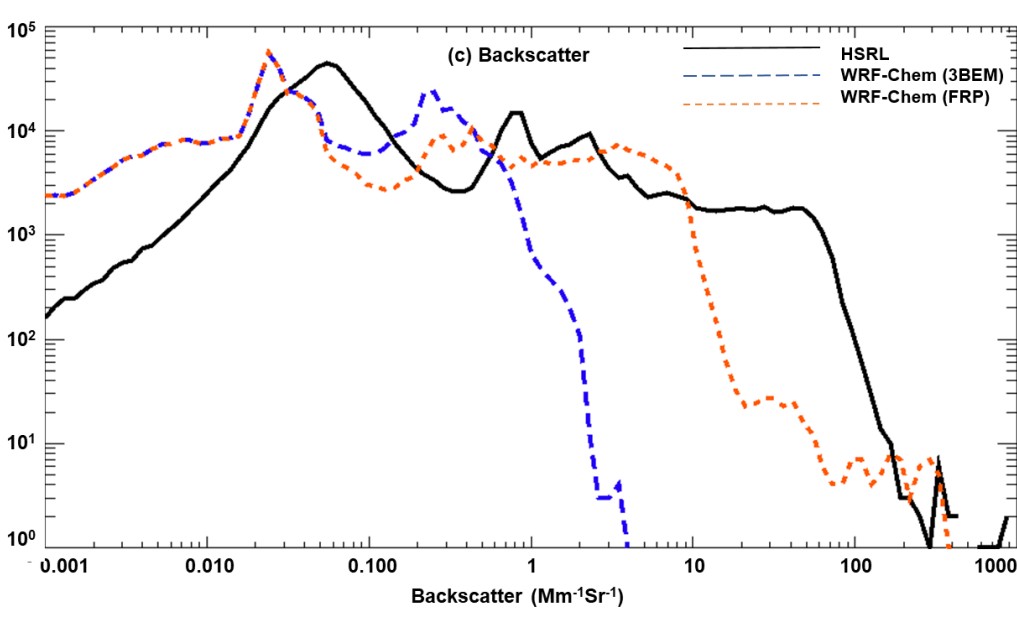



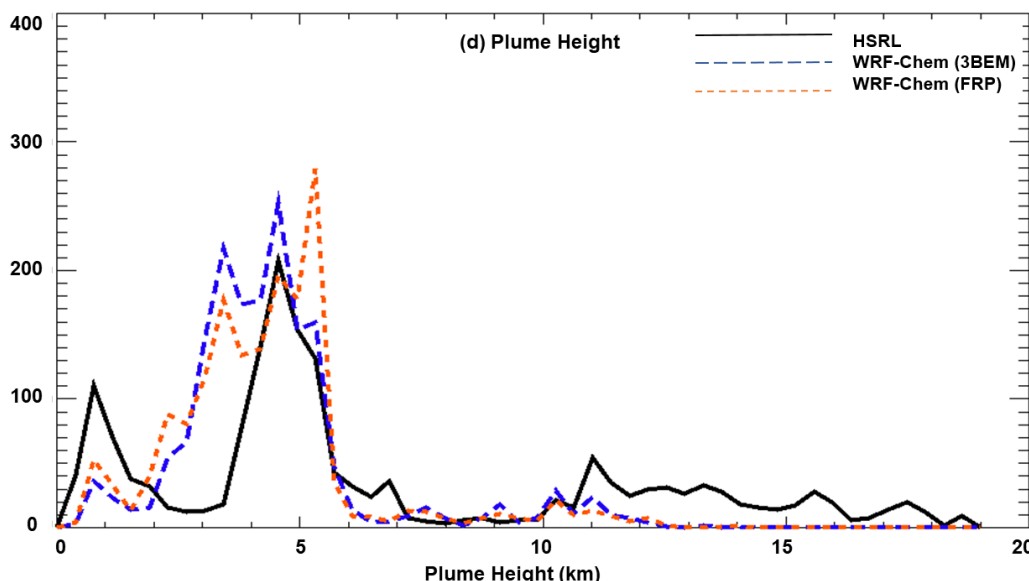


**Figure 15: Probability distribution functions for BC (a, top-left), OC (b, top-right), backscatter coefficient (c, bottom-left) and estimated plume heights (d, bottom- right). Note: BC and OC only represent in-plume cases.**


## 4.3. Statistical Comparison of WRF-Chem and FIREX-AQ Measurements (BC and OC)


**Table 1: Statistics for BC and OC (3BEM, FRP)**

| Flight | Average Bias | | Root Mean Squared Error | |
|---|---|---|---|---|
| | **3BEM** | **FRP** | **3BEM** | **FRP** |
| **BC** | | | | |
| August 3, 2019 | -1568 | -182 | 4676 | 4594 |
| August 6, 2019 | -319 | 271 | 1110 | 1349 |
| August 7, 2019 | -2525 | -1294 | 5675 | 4075 |
| **OC** | | | | |
| August 3, 2019 | -97 | -75 | 338 | 330 |
| August 6, 2019 | -44 | -36 | 185 | 181 |
| August 7, 2019 | -209 | -191 | 497 | 471 |




Table 1 shows statistical metrics of comparisons between the WRF-Chem simulated BC and OC
and the SP2 and AMS observations for the respective species for all FIREX-AQ DC-8 flights
considered in this work. The statistics reported are:
1.) $Average\ bias\ (MAB) = \left(\frac{1}{N}\right)\sum_{i=1}^{N}\left(X_{WRF-Chem_i} - X_{Obs_i}\right)$
2.) $Root\ Mean\ Squared\ Error\ (RMSE) = \sqrt{\left(\frac{1}{N}\right)\sum_{i=1}^{N}\left(X_{WRF-Chem_i} - X_{Obs_i}\right)^2}$
For BC, the 3BEM version had a low bias which was reduced significantly in the WRF-Chem FRP
version. However, the model was still underestimating BC as indicated by the negative MAB
values. For the August 6th flight, the WRF-Chem FRP version had a positive MAB which could
be due to the significant overestimation of BC during the low level smoke sampling period (Figure
10 (b) 19-20Z). This also contributes to the higher RMSE for the FRP version. For OC, the model
performance improved across all flights with a significant reduction in the MAB and lower RMSE
values than the 3BEM version. The improvements in model simulated aerosols were offset by the
inability of the model to simulate the aged part of the Williams Flats fire.

## 5. Conclusions



This study employs the Weather Research and Forecasting with Chemistry (WRF-Chem) model
(retrospective simulations) with GOES-16 FRP based methodologies to estimate wildfire
emissions, simulate wildfire plumerise and diurnal cycles to interpret in-situ and remote-sensing
measurements collected aboard the NASA DC-8 aircraft during the 2019 NASA-NOAA FIREX-
AQ field campaign and perform model evaluations. The primary focus is on the August 3-7, 2019,





science flights that sampled the Williams Flats fire in Washington. Main conclusions from this
evaluation are as follows:
1.) The FIREX-AQ observations were characterized by a variety of aerosol loading environments
which resulted in a large range of BC/OC and aerosol backscatter values during the August 3-8
science flights. These environments included fresh and aged smoke from Williams Flats and high-
altitude remnants of a plume that could have undergone long-range transport. The altitudes of
sampled smoke ranged from low-altitude (August 6) to a Pyro-Cb (August 8).
2.) The GOES-16 FRP based emissions employing the HRRR-Smoke methodology are
substantially higher than the standard emissions inventory (Freitas et al., 2011) in WRF-Chem
v3.5.1.
3.) Wildfire emissions in the standard WRF-Chem (3BEM version) resulted in significant
underestimation of carbonaceous aerosol (BC and OC) concentrations observed during the
Williams Flats sampling flights in FIREX-AQ. The implementation of FRP based emissions
improved the model simulation of BC and OC concentrations when compared to in-situ BC and
OC measurements. The FRP based modifications improved the capability of WRF-Chem in
simulating the high BC and OC enhancements observed during large wildfire events like the
Williams Flats fire.
4.) The simulated plume heights in the WRF-Chem FRP version did not show as large of changes
as the emissions. The HRRR-Smoke FRP-based plume-rise methodology produced similar plume
height distributions to the standard plumerise approach included in WRF-Chem v3.5.1 (Freitas et
al., 2007;2010). However, subtle differences were found during the flights considered. The aged


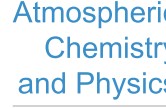

Williams Flats plume in Montana was not distinctively simulated (August 7 flight) while the plume
heights were lower for the Horsefly fire on August 6.
5.) The diurnal cycle imposed on wildfire emissions in WRF-Chem was also an important factor.
For multiple flights, the standard WRF-Chem v3.5.1 with a diurnal cycle peaking at 18UTC
(Freitas et al., 2011) simulated declining emissions, AOD, and BC and OC concentrations during
the latter stages of the science flights-while observations often showed increases during these
periods. This shortcoming was not found in the FRP-version which employed new FRP based
diurnal cycle functions which accounted for the variation with longitude during FIREX-AQ.
6.) WRF-Chem with the simplified GOCART mechanism could not adequately reproduce the
aerosol concentrations in the aged smoke (1 day of more of aging). This was observed for all
science flights that sampled aged smoke from Williams Flats. The potential reasons for this could
be biases in the aerosol dynamics (simulation of aerosol loss processes/transport) or chemistry
(e.g., no SOA in GOCART). It would be worthwhile to evaluate these flights in the future with a
more comprehensive chemistry mechanism (including SOA) to better understand the underlying
causes.
Overall, the HRRR-Smoke FRP based methodologies resulted in significant improvements in the
WRF-Chem forecasts for large wildfire events like the Williams Flats fire. These improvements
could translate into better estimates of impacts of large wildfire events on human health, which is
cause of concern given the current/future trends in wildfire activity in the US. These comparisons
also demonstrate that the FRP based emissions improve the forecast capability during major fire
events and would be useful to be incorporated in computational models providing air quality
forecasts.




**Author Contributions:** RBP conceptualized, supervised the study and developed the FRP based diurnal cycle functions. AK did the PREP-Chem (emissions), WRF-Chem (plumerise) development and carried out the WRF-Chem simulations. RBP and AK analyzed the FIREX-AQ and WRF-Chem data. AK wrote the manuscript draft with contributions from the co-authors. RA, GP, SF and GG developed the original HRRR-Smoke methodologies. CS provided the GOES-16 data. AL helped with setting up the WRF-Chem simulations. JPS, AEP, JMK provided the SP2-BC and fire flags data. JH provided the HSRL data. JLJ, PCJ and HG provided the AMS-OC data.

**Code/Data Availability:** FIREX-AQ measurements are available at: https://doi.org/10.5067/ASDC/FIREXAQ_Aerosol_AircraftInSitu_DC8_Data_1). The HSRL data are available at: https://doi.org/10.5067/ASDC/FIREXAQ_HSRL_AircraftRemoteSensing_DC8_Data_1).

**Competing Interests:** The authors declare that they have no conflict of interest.

**Acknowledgements:** We acknowledge funding support from the NOAA CPO AC4 grant. We would like to thank the FIREX-AQ leadership, the FIREX-AQ Science Team and the flight crews for their contributions towards the success of the campaign.



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
