# Peer review of "Simulating Wildfire Emissions and Plumerise using"

_Atmospheric Chemistry and Physics, 2022_

## Referee Comment (RC2)

Review of "Simulating Wildfire Emissions and Plumerise using Geostationary Satellite Fire Radiative Power Measurements: A Case Study of the 2019 Williams Flats fire"

Submitted to ACP

**Summary**

Accurately forecasting wildfire smoke concentrations with atmospheric models is challenging. Efforts to improve the performance of these tools, such as the analysis described in this paper, are important for helping air quality agencies effectively communicate warnings to the public ahead of severe smoke conditions. My comments highlight a few specific areas where the paper could be improved.

**General Comments:**

- More references to previous work would be helpful throughout, particularly in the Methods section when specific models/schemes are mentioned.
- Some of the discussion of FRP-based emission estimation approaches like GFAS and how they relate to more biogeochemically comprehensive datasets like GFED needs to be adjusted. The current presentation of these relationships is unclear/misleading. Please see specific comments on lines 117 and 120-122 below.
- A comparison of the emission factors used in the Freitas et al 2011 paper, the FRP approach described in this work, and more contemporary factors (e.g. SERA) would strengthen this paper and make it more broadly useful. Some other fire emissions systems use more recent emission factor estimates, so it would be helpful to understand if incorporating the FRP approach used here would yield similar improvement in those cases, or if much of the improvement observed in this work is in fact due to updated EFs compared to the 3BEM approach.

**Specific Comments:**

Lines 88-90. This sentence is a bit vague, could you perhaps be more specific about what aspect of wildfire regimes are you referring to here? E.g. changes in land cover/fuel loading, changes in ignition patterns due to expansion of the wildland-urban interface, etc

Line 106. It seems relevant to mention the representation of other meteorological variables here as well, e.g. wind direction.

Line 117. It seems important to point out here that the GFAS system relies heavily on relationships between FRP and land cover-dependent biomass consumption rates derived from the GFED dataset (see Kaiser et al 2012 Table 2 and discussion in section 2.3). Fundamentally you still need some estimate of fire size/location x fuel available x fuel consumed x emission factor to get an estimate of fire emissions. FRP-based approaches leverage relationships between some of these variables built from existing datasets to quickly combine some of these steps rather than calculating them explicitly.

Line 120-122. Again, this is a misleading assertion since FRP based approaches are built on existing datasets with more comprehensive biogeochemical modeling like GFED in order to calculate emissions per detection quickly. It would be more accurate to say something like "FRP based approaches like GFAS

are able to leverage key relationships, e.g. land cover specific consumption rates, from more comprehensive biogeochemical datasets like GFED in near-real time".

Line 191-198. It would be helpful to have more references in this section, for example is there a reference paper specifically for development of the "Arakawa Staggered C-Grid" mentioned on line 191?

Line 204. It seems overly specific to associate the model configuration you're using with a particular institution (even if that's where you're running the model), is there something unique but internally consistent about how the University of Wisconsin Madison group runs WRF-Chem compared to everyone else? Is this the official citation? I just haven't seen this done before.

Line 221. Is AOD actually calculated at 550 nm in the radiative transfer code? Or is it calculated at several other wavelengths and then interpolated? My understanding is the more common approach is to interpolate to 550, just clarifying.

Line 222. Please specify, how is hygroscopic growth accounted for? Lookup tables? Is there a reference for the approach?

Lines 225-226. Please include references for these models/schemes.

Line 239. Please provide relevant references demonstrating where in the literature these mechanisms have been evaluated.

Line 280. Can you specify what type of analyses? E.g. observational data from nearby monitors?

Lines 288-292. It might make sense to make the description of the diurnal functions its own section here and go into a bit more detail about how this was done.

Lines 304-305. Did this sentence get cut off? The "2 MODIS" is confusing, 2 what from MODIS? The two sensors on Aqua and Terra?

Line 455-457. It looks like the FRP based approach just generated higher emissions in general, which could be due to any number of factors. It's difficult to say since the values are shown on a log scale, but it looks like the relative change between August 3 and August 8 was actually larger in the 3BEM approach? I'm not sure it's appropriate to draw conclusions about the sensitivity of one approach vs. another to variability in fire behavior based solely on this figure, but perhaps I'm missing something.

Line 636. I recommend softening the interpretation of when things are "improved" or not in the FRP approach, here and throughout - it also looks like there are places where the FRP approach overestimates smoke emissions, so there may be an element of "right for the wrong reasons" in some cases. Trying to identify the specific sub-component of the estimate, e.g. fire size, fire location, fire timing, fire intensity, type of fuel, fuel moisture content, emission factor used, etc is critical for this type of exercise because otherwise you can scale emissions up or down to get "better" performance without knowing if the representation is more accurate at a process level.

Line 966. I'm not following the importance of the statements after "However" in Conclusion #4. In my view the main conclusion from the plume height comparisons was that use of the two different emission schemes didn't substantially alter the plume height representation, indicating that in these cases at least plume height representation in the model wasn't a central factor in the difference in performance between the two approaches compared to the flight data.

Line 975. It might be worth noting that conclusion 6 seems to be more of a second-order issue – other studies have shown that a simple scaling of ~1.5-2 on the OC allows for a decent representation of aerosol mass in many cases. The big first order issues relevant for modeling smoke transport seem to be more related to some of the other variables explored in this work, such as capturing fire size/location/timing, application of specific emission factors based on a variety of characteristics, and how those inputs interact with the representation of local meteorology.

**Technical Corrections:**

Line 218. Please explain what OA is (I didn't see where the term was introduced).

Line 504. I think you defined AOD above, don't need to redefine here

---

## Author Response (AR1)

**Responses to Reviewers**

We would like to thank both reviewers for their comments and suggestions. Please find our responses below.

**Reviewer 1**

**Comment**: This is a very interesting study to combine the WRF-Chem model and GOES-16 FRP-based emission and to comprehensively compare the model results with FIREX-AQ field campaign. By conducting the simulation based on the default emission module and newly-developed emissions, the authors provided a detailed analysis to interpret aerosol observations during the 2019 Williams Flats fire in Washington. Overall, I enjoy reading the manuscript, especially the introduction part which provides a really nice overview of the current emission inventory and model development for wildfire simulation. The experiments are well designed and the presented results are generally convinced. The topic is suitable for publication in ACP after addressing some specific comments listed below.

**Response**: Thanks very much for your encouraging comments!

**Comment**: It seems that the authors used the weather forecast data GFS to drive the WRF model. How did the WRF model reproduce the meteorological parameters during the wildfire, like wind field and air temperature stratification, which are vital for plume rise and the dispersion and transport of smoke? There have been many studies demonstrating that intense fire pollution would greatly modify the weather pattern. I wonder if the initial/boundary conditions from forecasted GFS data can well capture the evolution of the weather pattern and meteorological conditions on both local and regional scales. Therefore, I recommend more evaluation on the model performance of meteorology during the fire.

**Response**: This is a very good point. However, the scope of this study is limited to comparisons between the default WRF-Chem emissions and plumerise mechanisms and the modified FRP-based mechanisms employed in the HRRR-Smoke model. Although meteorology would play a very important role no doubt, we are solely focusing on the impacts the different methodologies would have on model performance. Thus, we are using the same meteorology in both 3BEM and FRP versions compared in the paper. Also, the impact of fires is not coupled to meteorology in the model. Therefore, these comparisons would be out of scope of the current work and could definitely form a part of a separate study. Hence, we would request a re-consideration of this comment for inclusion in the manuscript. Nevertheless, we have included comparisons of the WRF-Chem meteorology (horizontal wind speed and potential temperature) and the DC-8

meteorological measurements during the 3 flights (August 3, 6 and 7) evaluated in the paper. Please find the figures provided below.

[Figure]

**Figure 1: Comparisons of WRF-Chem meteorological variables (wind speed (a-c) and potential temperature) with the DC-8 measurements for the August 3, August 6 and August 7 FIREX-AQ DC-8 science flights**

**Comment**: Another, the authors conducted a great deal of work on the evaluations of AOD, BC and OC concentrations among different simulations. Regarding the AOD comparison, the satellite-detected AOD image could be employed to illustrate the distribution patterns in the real world.

**Response**: We have added a new section in the manuscript comparing the WRF-Chem simulated AOD (3BEM and FRP versions) with that from the GOES-16 and GOES-17 satellites. We have also derived AOD estimates from the HSRL backscatter observations. Please see Section 4.5.1 in the manuscript.

**Comment**: In addition to elaborating the disparities of WRF-Chem simulation using 3BEM and FRP emissions, I do think that the uncertainties in measurements ought to be briefly introduced. For instance, SP2 measurements tend to underestimate the BC concentration and it is somehow different from BC simulated in the model.

**Response**: We have added details and references about the uncertainty in the respective sections for SP2 (Section 3.2.1), AMS (Section 3.2.2) and HSRL (Section 3.2.3). In addition, regarding your specific comment on SP2, we now include an explicit uncertainty estimate for the SP2-determined rBC concentrations (<=40%). We had already addressed the corrections applied to address the fact that SP2 does not quantify the entire range of rBC size; we have expanded that explanation to address the concern in the model comparison. In short, the SP2 measurements were scaled (approximately 10%) to represent total accumulation-mode rBC concentrations. The GOCART model scheme deals only with bulk black carbon, without size resolving it, so discrepancies in the comparison between the model and measurements are only relevant to the small extent that the inventories include non-accumulation-mode emissions.

**Comment:** Another suggestion is to briefly describe the Williams Flats fire, air quality observations and the flight measurements before comparing the model results in detail.

**Response**: We have described the Williams Flats fire in Section 4.1 in the manuscript. The air quality observations and flight measurements are described in Section 3.2.

**Comment**: I am a little bit missing in Section 4.2, especially when the titles of each subsection are flight date.

**Response**: We have re-arranged the sections in the manuscript and re-named them.

**Comment**: More information concerning the fire spots and flight date could be plotted on Figure 1.

**Response**: We have added additional information in Figure 1. However, this figure was included just to provide a macroscopic overview of the area of focus of the campaign. The details of the flights are discussed extensively in Sections 4.1 and 4.2 and in the discussion of the results, while the fire spots are plotted in the AOD figures (Figure 7).

**Comment:** One may get a clearer picture if some general descriptions for each case are added before 4.2.

**Response:** We have added a new section (Section 4.2) in the manuscript which provides a general description of the individual flights before discussing the results.

**Comments:** Another, the set of figures (Fig.4-7, Fig. 8-11...) for each flight are quite similar, which are suggested to be diversified according to the main points from each comparison.

**Response:** We have re-organized the sections, re-done the figures and re-named them in order to reduce the monotonicity. Please see Section 4.4 onward.

**Minor comments**:

**Comment**: The authors used the units µg/sm3 throughout the manuscript. Please double check sm3.

**Response**: "s" in sm$^{-3}$ here represents standard temperature and pressure (STP).

**Comment**: Figure 2, The time is suggested to be presented as UTC since UTC time is used throughout the manuscript.

**Response**: Done.

**Comment**: Line 975-981: Since that GOCART does not resolve the size distribution of BC, OC and other secondary aerosol components, as mentioned in Line 217-219, a more comprehensive chemistry mechanism with size distribution treatment is also needed.

**Response**: Done.

**Comment**: Table 1: add the units for BC and OC concentration

**Response**: We have substituted Table 1 with a figure (Figure 12) and added the units.

**Reviewer 2**

**General Comments:**

- **Comment**: More references to previous work would be helpful throughout, particularly in the Methods section when specific models/schemes are mentioned.

  **Response**: We have added more references.

- **Comment**: Some of the discussion of FRP-based emission estimation approaches like GFAS and how they relate to more biogeochemically comprehensive datasets like GFED needs to be adjusted. The current presentation of these relationships is unclear/misleading. Please see specific comments on lines 117 and 120-122 below.

  **Response**: Done (Please see responses below).

- **Comment**: A comparison of the emission factors used in the Freitas et al 2011 paper, the FRP approach described in this work, and more contemporary factors (e.g. SERA) would strengthen this paper and make it more broadly useful. Some other fire emissions systems use more recent emission factor estimates, so it would be helpful to understand if incorporating the FRP approach used here would yield similar improvement in those cases, or if much of the improvement observed in this work is in fact due to updated EFs compared to the 3BEM approach.

  **Response**: We have used the same emission factors in both the 3BEM and FRP versions in order to make sure the changes only represent the differences in the emissions estimation methodologies. So, even if we use emission factors from other sources, the changes should remain the same. However, the magnitude of emissions for each version would change. We have added a paragraph in the text stating this:

  *"We used the same emission factors in both the 3BEM and FRP versions to ensure that the changes in emissions solely represent the differences in the two methodologies. Considerable progress has been made in improving upon the emission factor estimates used in this study. For example, subsequent work by Akagi et al. 2011 (referred to as AK11), and Andreae 2019 (referred to as AN19) have resulted in new emission factor estimates for biomass burning. In comparison to these studies, our OC emission factors for tropical forests were 9% higher than*

*AK11 (BC: 21%) and 15% higher than AN19 (BC: 23%) while for extratropical forests the emission factors were the same as AK11. AN19 did not report emission factors for extratropical forests. For savanna/grasslands, OC emission factors were 18% higher than AK11 (BC: 20%) and 6% higher than AN19 (BC: 15%). Thus, incorporation of these emission factors could alter the magnitude of emission estimates (for both 3BEM and FRP versions) reported in Figure 4."*

**Specific Comments:**

- **Comment**: Lines 88-90. This sentence is a bit vague, could you perhaps be more specific about what aspect of wildfire regimes are you referring to here? E.g. changes in land cover/fuel loading, changes in ignition patterns due to expansion of the wildland-urban interface, etc.

  **Response**: We have clarified the sentence as *"Wildfire regimes (e.g. frequency, size and severity) have altered significantly over the past few years in the United States (US) with climate change hypothesized to be a major driving force."*

- **Comment**: Line 106. It seems relevant to mention the representation of other meteorological variables here as well, e.g. wind direction.

  **Response**: We have changed the sentence to *"The ability of computational models to accurately simulate air quality impacts during wildfire events is critically dependent on the inputs such as the estimated emissions, the simulated altitude of the emissions (smoke injection height, or plume-rise) (Val Martin et al., 2012;Carter et al., 2020) and meteorological variables (e.g. wind direction)."*

- **Comment**: Line 117. It seems important to point out here that the GFAS system relies heavily on relationships between FRP and land cover-dependent biomass consumption rates derived from the GFED dataset (see Kaiser et al 2012 Table 2 and discussion in section 2.3). Fundamentally you still need some estimate of fire size/location x fuel available x fuel consumed x emission factor to get an estimate of fire emissions. FRP-based approaches leverage relationships between some of these variables built from existing datasets to quickly combine some of these steps rather than calculating them explicitly.

  **Response**: We have modified the text as mentioned in the response below.

- **Comment**: Line 120-122. Again, this is a misleading assertion since FRP based approaches are built on existing datasets with more comprehensive biogeochemical modeling like GFED in order to calculate emissions per detection quickly. It would be more accurate to say something like "FRP based approaches like GFAS are able to leverage key relationships, e.g.

land cover specific consumption rates, from more comprehensive biogeochemical datasets like GFED in near-real time".

**Response**: We have modified the text as: *"These approaches combine FRP measurements with biomass burned rates to estimate emissions. A major advantage FRP based approaches like GFAS provide is the ability to leverage key relationships, e.g. land cover specific consumption rates, from more comprehensive biogeochemical datasets like GFED in near-real time."*

- **Comment**: Line 191-198. It would be helpful to have more references in this section, for example is there a reference paper specifically for development of the "Arakawa Staggered C-Grid" mentioned on line 191?

  **Response**: We have added the details in the text as*: "It uses the Arakawa Staggered C-Grid horizontally whereas the vertical levels in the model are defined using a terrain following sigma-hybrid coordinate system (Skamarock et al., 2019) [Section 3.2 and Section 1.2], Arakawa and Lamb 1977"*.

- **Comment**: Line 204. It seems overly specific to associate the model configuration you're using with a particular institution (even if that's where you're running the model), is there something unique but internally consistent about how the University of Wisconsin Madison group runs WRF-Chem compared to everyone else? Is this the official citation? I just haven't seen this done before.

  **Response**: Yes, we agree that it sounds a bit too specific. However, this was done since the configuration of WRF-Chem as used at the University of Wisconsin Madison is different than WRF-Chem used at other institutions. For example, WRF-Chem at the University of Iowa uses BCs from WACCM while we use them from RAQMS. Similarly, WRF-Chem at NCAR uses the FINN emissions, WRF-Chem at UCLA uses QFED, while we use 3BEM. Hence, it was important to clarify that the model results are obtained using a specific WRF-Chem configuration run at the University of Wisconsin Madison. The terminology of associating an institution name with their WRF-Chem version has been previously used by Ye et al. 2021 "Evaluation and intercomparison of wildfire smoke forecasts from multiple modeling systems for the 2019 Williams Flats fire" (https://acp.copernicus.org/articles/21/14427/2021/). We have minimized the use of this term in the manuscript.

- **Comment**: Line 221. Is AOD actually calculated at 550 nm in the radiative transfer code? Or is it calculated at several other wavelengths and then interpolated? My understanding is the more common approach is to interpolate to 550, just clarifying.

**Response**: We have added the following in the text: *"The Aerosol Optical Depth (AOD) in the model is calculated at 550 nm by vertical integration of the aerosol extinction using Mie scattering based look-up tables of effective radius and extinction coefficients as a function of relative humidity."*

- **Comment**: Line 222. Please specify, how is hygroscopic growth accounted for? Lookup tables? Is there a reference for the approach?

  **Response**: We have added the following sentence in the text *"Hygroscopic growth is accounted for by determining hydroscopic growth factors from look-up tables computed using Mie theory following Martin et al. (2003) and extinction efficiencies are used as a function of mole fraction."*

- **Comment**: Lines 225-226. Please include references for these models/schemes.

  **Response**: We have added more references (Lines 201-208).

- **Comment**: Line 239. Please provide relevant references demonstrating where in the literature these mechanisms have been evaluated.

  **Response**: We have added the following references in the text:

  "*It also includes comprehensive stratospheric and tropospheric chemistry mechanisms (Pierce et al., 2007), which have been extensively evaluated (Kiley et al., 2003;Fairlie et al., 2007;Pierce et al., 2009;Al-Saadi et al., 2008;Natarajan et al., 2012;Yates et al., 2013;Sullivan et al., 2015;Baylon et al., 2016;Huang et al., 2017).*"

- **Comment**: Line 280. Can you specify what type of analyses? E.g. observational data from nearby monitors?

  **Response**: We have added the following in the text: "*In retrospective mode, the model has the same configuration as the forecast mode except that fire detections are for the current day, and the NOAA National Center for Environmental Prediction (NCEP) Global Data Assimilation System (GDAS) (Wang and Lei, 2014) is used for initial and lateral boundary meteorological conditions and RAQMS is used for initial and lateral boundary aerosol conditions.*"

- **Comment**: Lines 288-292. It might make sense to make the description of the diurnal functions its own section here and go into a bit more detail about how this was done.

  **Response**: We have added more details regarding the development of the diurnal cycle functions. "*The default diurnal cycle function for biomass burning emissions in WRF-Chem is a Gaussian function peaking at 18UTC (Freitas et al 2011). The GOES-16 FRP measurements during the FIREX-AQ period (August – September 2019) were divided into three zones based on longitude (zone 1 (blue in Figure 2): -130W to -110W, zone 2 (green in Figure 2): -110W to -90W and zone 3 (red in Figure 2): -90W to -70W) and the mean FRP diurnal profiles were constructed for each zone. The default diurnal cycle function used in WRF-Chem was iteratively adjusted to match the FRP profiles for each zone resulting in three diurnal cycle functions. These diurnal functions were used in the FRP version.*"

- **Comment**: Lines 304-305. Did this sentence get cut off? The "2 MODIS" is confusing, 2 what from MODIS? The two sensors on Aqua and Terra?

  **Response**: We have clarified the sentence further as "*The model uses fire location (latitude, longitude) and FRP measurements from 4 polar orbiting satellites, 2 (Suomi-NPP and NOAA-20) for VIIRS (375m resolution I-band Active Fire (AF) algorithm which is based on the Moderate Resolution Imaging Spectroradiometer (MODIS) Collection 6 retrieval (Giglio et al., 2016)) and 2 (Terra and Aqua) for MODIS.*"

- **Comment**: Line 455-457. It looks like the FRP based approach just generated higher emissions in general, which could be due to any number of factors. It's difficult to say since the values are shown on a log scale, but it looks like the relative change between August 3 and August 8 was actually larger in the 3BEM approach? I'm not sure it's appropriate to draw conclusions about the sensitivity of one approach vs. another to variability in fire behavior based solely on this figure, but perhaps I'm missing something.

  **Response**: Yes, emissions using the FRP based approach are higher than 3BEM. This is consistent with the findings of Ye et al. 2021 who found FRP based approaches were higher than the burned area based approaches. This is mentioned in the text (Section 4.3).
  Emissions changed 335% (OC), 286% (BC) for the 3BEM approach and 56% (OC), 49% (BC) for the FRP approach. That part of the text was referring to the magnitude of emissions change rather than the rate. Regarding the sensitivity of the two methods, we agree that additional analysis would better support that statement. That analysis would potentially require a continuous study of the fire as it evolves accompanied by comparisons of the changes in emissions. It could form part of a separate study. So, we have removed that part of the text.

- **Comment**: Line 636. I recommend softening the interpretation of when things are "improved" or not in the FRP approach, here and throughout - it also looks like there are places where the FRP approach overestimates smoke emissions, so there may be an element of "right for the wrong reasons" in some cases. Trying to identify the specific sub-component of the estimate, e.g. fire size, fire location, fire timing, fire intensity, type of fuel, fuel moisture content, emission factor used, etc. is critical for this type of exercise because otherwise you can scale emissions up or down to get "better" performance without knowing if the representation is more accurate at a process level.

  **Response**: We have minimized the use of the term "improve". We have substituted it with "better agreement with observations".

- **Comment**: Line 966. I'm not following the importance of the statements after "However" in Conclusion #4. In my view the main conclusion from the plume height comparisons was that use of the two different emission schemes didn't substantially alter the plume height representation, indicating that in these cases at least plume height representation in the model wasn't a central factor in the difference in performance between the two approaches compared to the flight data.

  **Response**: Those statements were briefly mentioning the differences observed in estimated plume heights in the two methodologies. However, we agree with your synopsis. We have removed the statements after "However". We have modified the statement to: *"The simulated plume heights in the WRF-Chem FRP version did not show as large of changes as the emissions. The HRRR-Smoke FRP-based plume-rise methodology produced similar plume height distributions to the standard plumerise approach included in WRF-Chem v3.5.1 (Freitas et al., 2007;2010). Thus, the better performance of the WRF-Chem FRP version was mainly driven by the higher emissions in the FRP-based version."*

- **Comment**: Line 975. It might be worth noting that conclusion 6 seems to be more of a second-order issue – other studies have shown that a simple scaling of ~1.5-2 on the OC allows for a decent representation of aerosol mass in many cases. The big first order issues relevant for modeling smoke transport seem to be more related to some of the other variables explored in this work, such as capturing fire size/location/timing, application of specific emission factors based on a variety of characteristics, and how those inputs interact with the representation of local meteorology.

  **Response**: We have modified conclusion #6 and added: *"In addition to the primary factors such as emissions, plume-height and wildfire diurnal cycle estimation, second-order issues like"*.

**Technical Corrections:**

- **Comment**: Line 218. Please explain what OA is (I didn't see where the term was introduced).

  **Response**: Done.

- **Comment**: Line 504. I think you defined AOD above, don't need to redefine her

  **Response**: Done.

---

## Author Response (AR2)

**Responses to Reviewers**

**Comment:** The title page of *pdf. manuscript file must include the full institutional addresses of all authors. However, country name is missing from the affiliations. Please, add it for the next revision.

**Response:** We have added the country names.

**Comment:** Regarding your figure #8: please ensure that the colour schemes used in your maps and charts allow readers with colour vision deficiencies to correctly interpret your findings. Please check your figures using the Coblis – Color Blindness Simulator (https://www.color-blindness.com/coblis-color-blindness-simulator/) and revise the colour schemes accordingly.

**Response:** We have revised the color schemes in the figure.

**Comment:** Regarding your figure #1: with the next revision, please add the copyright icon as follows: "Image: © Google Earth" to the caption of this figure.

**Response:** We have added it.

**Comment:** Line 489 : provide the links for Inciweb or cite it as a reference.

**Response:** Unfortunately, the Inciweb links have expired. Therefore, we have inserted links from alternate sources that provide the same information. Please see the following:

Lines 486-488: *"The 100% containment date for the fire was reported to be August 25, 2019, and it burned an estimated 44,446 Acres (https://data.statesmanjournal.com/fires/incident/6493/williams-flats-fire/)"*

Lines 493-495: *"The Horsefly fire started on August 5, 2019, 15 miles east of Lincoln in the Lewis and Clark County (Montana) and burned 1274 acres in the first 24 hours (https://data.redding.com/fires/incident/6502/horsefly-fire/)."*

Lines 495-497: *"The fire was reported to have burned 1350 acres till August 23, 2019, with zero growth reported in the prior week (https://data.redding.com/fires/incident/6502/horsefly-fire/)."*

Also, the statement (Line 559-561 in the revised submission): *"The Williams Flats fire increased from 10,438 560 acres to 40,000 acres during August 3 -8 (source: Inciweb August 4 and August 9, 9:00 am update) which is reflected in the increase in BC and OC emissions."* has been changed

to *"The Williams Flats fire increased significantly in size during August 3 -8 (Ye et al. 2021)"* due to the Inciweb update link not being available (Lines 549-550 in the current version).

**Comment:** Figure 10: the labels are too small to be identified clearly. Other figures that have similar problems also need to be revised for clarity.

**Response:** We have reviewed the figures in the manuscript. We have increased the label size in all the figures and modified the figure captions (Figure 4 and Figure 9) for clarity. We have also re-verified the figure references in the text. There were a few cases where the figure references had to be updated so that they are pointing to the correct figure in the panels.

Lines 650, 652, 653, 665: References changed to Figure 7 instead of Figure 4

Line 841: Figure (10 (a), (b)) to Figure (10 (a), (d))

Line 865: Figure (10 (d), (e)) to Figure (10(b), (e))

Line 887: Figure 10 ((g), (h)) to Figure 10 ((c), (f))

**Minor corrections**:

Line 136: Corrected the section number: "described in Section 3" to "described in Section 2"

Line 471: Corrected the sentence *"Ye et al. 2021 also reported the current inability of models to represent the simulate pyro-Cb events"* to *"Ye et al. 2021 also reported the current inability of models to simulate pyro-Cb events current inability of models to simulate pyro-Cb events"*